# Observer-Based H∞ Controller Design for High Frequency Stick-Slip Vibrations Mitigation in Drill-String of Rotary Drilling Systems

**Rami Riane [1], Mohamed Zinelabidine Doghmane [1,2], Madjid Kidouche [1] and Sofiane Djezzar [3,*]**

[1] Laboratoire d'Automatique Appliquée, Faculté des Hydrocarbures et de la Chimie, Université M'Hamed Bougara de Boumerdes, Boumerdes 35000, Algeria; r.riane@univ-boumerdes.dz (R.R.); m.doghmene@univ-boumerdes.dz (M.Z.D.); mkidouche@univ-boumerdes.dz (M.K.)

[2] Reservoirs Evaluation Department, OPE, Exploration Division, Sonatrach, Hassi Messaoud 30500, Algeria

[3] Petroleum Engineering Department, University of North Dakota (UND), Grand Forks, ND 58202, USA

[*] Correspondence: sdjezzar@undeerc.org

**Abstract:** The drilling process is among the most crucial steps in exploration and production activities in the petroleum industry. It consists of using mechanical mechanisms to crush rocks by the drill bit to pass through the different geological layers. The drill-string continuously transforms the rotational movement from the top drive motor to the drill bit through the drill pipes. Due to the strong interactions with the rocks, aggressive vibrations can arise in the drill-string in its three dimensions, and consequently, this may create three types of synchronous vibrations: axial, lateral, and torsional. The severe status of the latter is known as the stick-slip phenomenon, and is the most common in rotary drilling systems. Based on field observations, it has been inferred that the high frequency stick-slip vibrations may lead to drill-string fatigues and even to premature rupture. In the best case, it reduces the drilling efficiency by decreasing the rate of penetration, due to which the drilling operations become proportionally expensive. The main novelties of this research work are the design of an H∞ observer-based controller to mitigate the high frequency stick-slip vibrations, and the quantitative analysis of the vibrations' severity for ten degrees of freedom model. The observer is designed to estimate the non-measurable rotational velocity of the drill bit due to the severity of the vibrations, while the controller is dedicated to suppressing the vibrations by using the top drive inputs. Thus, many scenarios have been considered to test and analyze the observer performance and the controller robustness. Furthermore, a comparison with the LQG observer-based controller has been conducted, where H∞ has demonstrated better efficiency in suppressing the stick-slip vibrations under unstructured perturbations.

**Keywords:** petroleum industry; high frequency stick-slip vibrations; h∞ observer-based controller; rotary drilling systems; drill bit velocity





## 1. Introduction

The drilling process is the ensemble of all operations that allow the digging of holes in the earth either vertically, deviated, or even horizontally. The petroleum industry generally uses rotary drilling systems to ensure the production of hydrocarbons (Figure 1a). Since this process is one of the main essential parts of the well's total cost, its efficiency, reliability, and performance are very important features for deep well drilling. The occurrence of drill-string vibrations is one of the main restrictive factors of these drilling aspects because it reduces the borehole quality and generates premature wears of the drilling equipment. Moreover, in extreme cases, it encourages the drill-string breakage by holding the drilling operations. Therefore, mastering the dynamics of such vibrations and its suppression in a robust way have been attracting a lot of interest in the last few years by researchers due to its great economic outcomes in the petroleum industry [1]. To this end, the most common type

of vibration, known as stick-slip, has been considered in this research work. The stick-slip is primarily a torsional vibration generated by the drill-string elasticity and the discontinuous nature of the torque on bit (Tob) [2]. Even though the diameter of the borehole can be larger than the drill pipes' diameter, there will often be a direct contact between the drill-string and the borehole of the well. This is caused by two main reasons: the first is due to the bending of the long drill-string, which is caused by its weight and imbalance centrifuge forces, while the second reason is due to the fact that the well deviation has become more considerable due to deep target reservoirs. Hence, the frictional contact forces can be considerable enough to cause the stick-slip vibrations along the drill-string.

Aiming to overcome the repercussions of the stick-slip vibrations appearance during drilling, many contributions and approaches have been developed in the last few years. From a practical point of view, the drillers have adopted a method based on manual drilling parameters' manipulation by reducing the Wob and increasing the top drive rotational velocity (or torque). Even this procedure can free the drill bit from the stuck phase, it takes considerable time in which the borehole can face too many vibrations [3]. Thence, more reliable and robust automatic approaches have been developed and introduced as plug-in software solutions. More accurately, the soft torque rotary system (STRS) was first introduced in [4], and it is based on eliminating the stick-slip vibrations by the top drive torque manipulation in a closed-loop form. This solution was subsequently adopted by other companies under different names [5,6]. However, it necessitates a direct measurement of the drill bit velocity [7], which is not possible if the measurement while drilling (MWD) tool is not logged. From an academic point of view, several papers have been published about different control strategies for stick-slip vibrations mitigation in the drill-string [8,9], most of which have used a two degrees of freedom model of the rotary drilling system in the controller design [10–14]. This model has proven to be simple, practical, and sufficient, but only if the supposition given by [15], which indicates that stick-slip occurs at low frequencies, is verified. A model with several degrees of freedom has been proposed by [16], and a non-linear model by [6]. The latter works assumed that the drill bit angular velocity is measurable, which necessitated the implementation of MWD tools for real-time control purposes. More recent works in this topic proposed the use of observers for drill bit velocity (and torque) estimation [17], then stick-slip mitigation based on PI controller [18,19]. In these works, the top drive velocity and torque have been considered as the control input signals to mitigate the stick-slip vibrations in the drill-string. Even the observer estimations were acceptable, but the tuning process of the PI controller has influenced the controller robustness. Another approach based on automatic variation of the Wob has been adopted in [20] to eliminate these vibrations. However, its limitation is mainly due to the slow dynamic caused by the Wob variation in comparison to the dynamics of stick-slip vibrations [1,21].

This research work is focused on the estimation and control of drill bit velocity to suppress the high frequency stick-slip vibrations in a more robust way by using the H∞ observer-based controller. The main novelty of this work is that an observer is designed on the basis of a linear model to estimate the non-measurable states as well as the unknown input of ten degrees of freedom model subject to nonlinear input. In addition to that, the severity of the stick-slip vibrations has been quantitatively analyzed on the basis of the outcomes of the designed observer. A comparison with LQG observer-based controller has been conducted, for which H∞ has demonstrated better performance in suppressing the high-frequency stick-slip vibrations under the unstructured perturbations. The manuscript is organized as follows. In Section 2, the types of vibrations have been overviewed with some focus on stick-slip vibrations whose severity has been quantitatively classified. The open-loop model with ten degrees of freedom has been tested under different scenarios to validate the model reliability and its estimated drill bit velocity. In Section 3, the mathematical design of H∞ observer-based controller has been detailed for the drill-string model, where the estimation results allowed us to design and investigate the controller robustness. In Section 4, different scenarios have been introduced to test the proposed observer-based

controller against the high frequency stick-slip vibrations severities. Moreover, the proposed controller results have been compared with the LQG observer-based controller, for which H∞ has demonstrated better performance than LQG, especially toward the unstructured perturbations. Section 5 presents conclusions and recommendations for future research works.

## 2. Drill-String Model under Stick-Slip Vibrations

Based on [18], the vibrations can be generally classified into three categories: free, forced, and self-sustaining. In self-sustaining vibrations, a periodic force is regenerated, which then excites the vibrations themselves. Thus, if the system is prevented from vibrating, then the excitement force disappears [22]. On the other hand, for forced vibrations, the excitation force is independent of the vibrations, and it can persist even when the vibrations are suppressed [23]. Fortunately, the drilling process vibrations regime is self-sustaining, due to which a constant disturbance can lead to unstable oscillating regime, which in its turn evolves towards a stable limit cycle [24,25]. Therefore, the stick-slip vibrations themselves can generate their own excitement through the axial vibrations [26]. Hence, by eliminating the high-frequency stick-slip vibrations as quick as possible, we can prevent the generation of the other types of vibrations too [27].

### 2.1. Torsional Vibrations

The severe torsional vibrations manifest as the stick-slip phenomena, in which a cyclic drill bit stop can be noticed [7]. During the stick phase, the drill-string shown in Figure 1b, rotated by the top drive from the surface, is twisted due to its elasticity, while the drill bit will remain stuck until the cumulated bottom torque becomes greater than the static frictional torque [2]. The sudden drill bit sleep leads to abnormal drill-string acceleration, and it may even exceed the top drive velocity by several times [28]. Even though the nonlinear dynamic of stick-slip vibrations is poorly understood, most researchers have concurred that during this cyclic phenomenon, the drill bit velocity variations are synchronized with resistant torque variations [1,17]. This is mainly due to the difference between the static and dynamic frictions between the polycrystalline diamond compact (PDC) cutting edges and the rocks of the geological layer being drilled [29–32]. The mathematical equations of motion for the drill-string under stick-slip vibrations are generally established based on a torsional system with one or two degrees of freedom as given in [4,12,29,30,33]. It has been concluded through these research works that the torsional vibrations appear more when using the PDC drill bit because it consumes more torque than the tri-cone bit, for example [27]. Since the stick-slip is a self-sustaining phenomenon, it is necessary to develop a new approach for the reliable estimation, fast detection, and robust suppression of such harmonics. The new mathematical formula given by (1) has been proposed in this study to quantify the stick-slip vibrations' severity *SS*% [34].

$$SS\% = \frac{\Delta\,Speed}{Mean\,(Speed)} \tag{1}$$

The change in drill bit rotational velocity is normalized by its mean; the *SS*% is classified into four levels as detailed in Table 1. So, the elimination of this type of vibration will be based on its severity level.

**Table 1.** Stick-slip severity classification.

| Stick-Slip Severity Level | Classification | SS% |
|:---:|:---:|:---:|
| 0 | Verylow | 0 to 50% |
| 1 | Low | 50% to 100% |
| 2 | Mean | 100% to 150% |
| 3 | High | >150% |

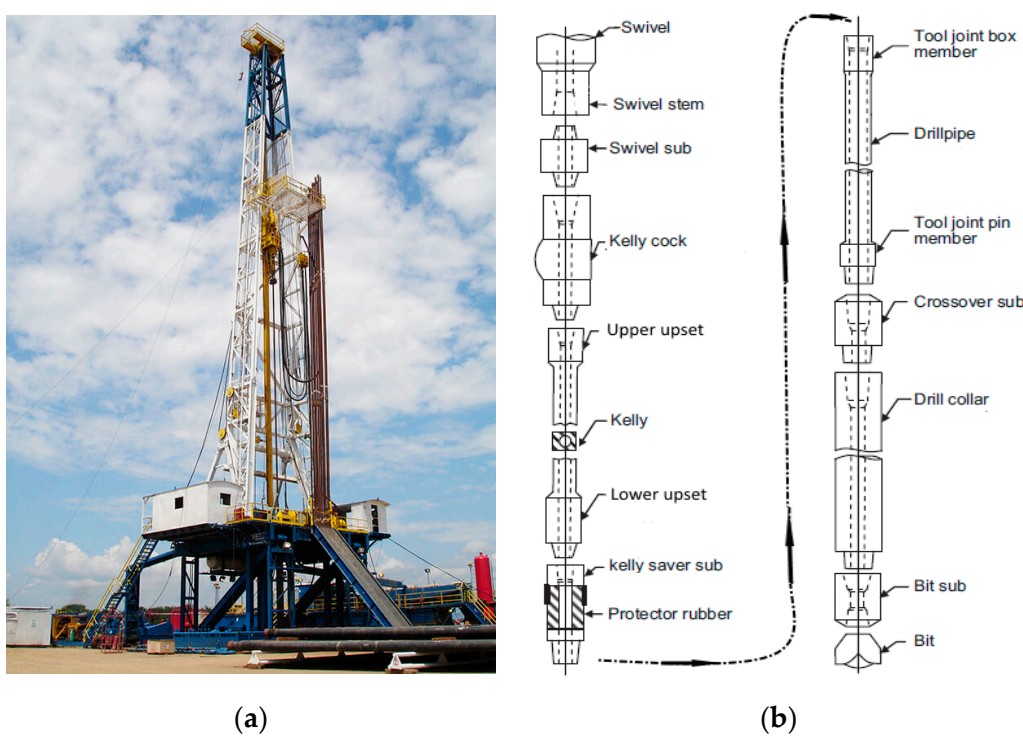

**(a)** **(b)**

**Figure 1.** Rotary drilling system: (**a**) Operational rig from Algeria, (**b**) Its drill-string scheme.

### 2.2. Drill-String Model

The drill-string model developed in this study is based on dividing the drill-string into ten parts connected in series, where each part is presented by a torsion pendulum subjected at its ends to viscous and dry frictions. The upper part is attached to the top drive, while the lower part is related to the borehole assembly (BHA), which is under viscous friction and the Tob. The mathematical model of the drill-string can be studied in two equations: the first equation is dedicated to the dynamic of the Top drive as given by Equation (2), while the second is dedicated to the BHA as given by Equation (3) [21].

$$
\begin{cases}
J_t \ddot{\varphi}_t = nKI - k\varnothing - C(\dot{\varphi}_t - \dot{\varphi}_b) - C_t \dot{\varphi}_t \\
L\dot{I} = U - RI - nK\dot{\varphi}_t
\end{cases}
\tag{2}
$$

$$
J_b \ddot{\varphi}_b = k\varnothing + C(\dot{\varphi}_t - \dot{\varphi}_b) - C_b \dot{\varphi}_b - Tob(\dot{\varphi}_b, \ Wob)
\tag{3}
$$

where $J_t$ is the equivalent moment of inertia, $\varphi_t$, $\dot{\varphi}_t$, $\ddot{\varphi}_t$ are the position, the angular velocity, and the acceleration of the upper part of the drill-string, $\varphi_b$, $\dot{\varphi}_b$, $\ddot{\varphi}_b$ are the position, the angular velocity, and the acceleration of the lower part of the drill-string (BHA), $C_t$ is the viscous friction coefficient, $k$ is the stiffness constant, $C$ is the viscous friction coefficient under torsion, $n$ is the torque transmission ratio of the gearbox, $K$ is the torque constant, $I$ is the current consumed by the electrical motor of the top drive, $U$ is the power supply voltage, and $R$ and $L$ are the armature resistance and inductance, respectively, with $\varnothing = \varphi_t - \varphi_b$. $J_b$, $C_b$ are respectively the equivalent moment of inertia and the viscous friction coefficient at the BHA. $Tob(\dot{\varphi}_b, \ Wob)$ is the unknown torque on bit, which is a function of the drill bit velocity and the weight on bit [35]. The rock-bit interaction block diagram, which includes the Tob, is shown in Figure 2. Equations (2) and (3) have been rewritten in the state space form given by (4).

$$
\begin{cases}
\dot{x}(t) = Ax(t) + B\Gamma(t) \\
y(t) = Cx(t)
\end{cases}
\tag{4}
$$

where $x(t) \epsilon \mathbb{R}^4$ is the states' vector, $\Gamma(t) \epsilon \mathbb{R}^2$ is the inputs' vector, and $y(t) \epsilon \mathbb{R}^2$ is the outputs' vector, with $x(t) = \begin{bmatrix} \varnothing & \dot{\varphi}_t & \dot{\varphi}_b & I \end{bmatrix}^T$, $\Gamma(t) = \begin{bmatrix} Tob & U \end{bmatrix}^T$, and $y(t) = \begin{bmatrix} \dot{\varphi}_t & I \end{bmatrix}^T$ [36].

$$A = \begin{pmatrix} 0 & 1 & -1 & 0 \\ -\frac{k}{J_t} & -\frac{(C_t+C)}{J_t} & \frac{C}{J_t} & \frac{nK}{J_t} \\ \frac{k}{J_b} & \frac{C}{J_b} & -\frac{(C_t+C)}{J_b} & 0 \\ 0 & -\frac{nK}{L} & 0 & -\frac{R}{L} \end{pmatrix}, \quad B = \begin{bmatrix} 0 & 0 \\ 0 & 0 \\ -\frac{1}{J_b} & 0 \\ 0 & \frac{1}{L} \end{bmatrix}, \quad C = \begin{bmatrix} 0 & 1 & 0 & 0 \\ 0 & 0 & 0 & 1 \end{bmatrix}$$

The mathematical model given by the previous matrix is for two degrees of freedom. However, in this study, we have used a ten-degree of freedom model given by [37] as illustrated in Figure 3.

### 2.3. Open Loop System Responses

The scenarios carried out in this subsection have the objective of testing the model reliability against different top drive input torque signals. In addition, it is a crucial step to investigate and validate the effectiveness of the H∞ observer before designing the controller in cascaded structure [38,39]. The input variations were mainly based on drillers' recommendations extracted from practical drilling experiences [40,41].

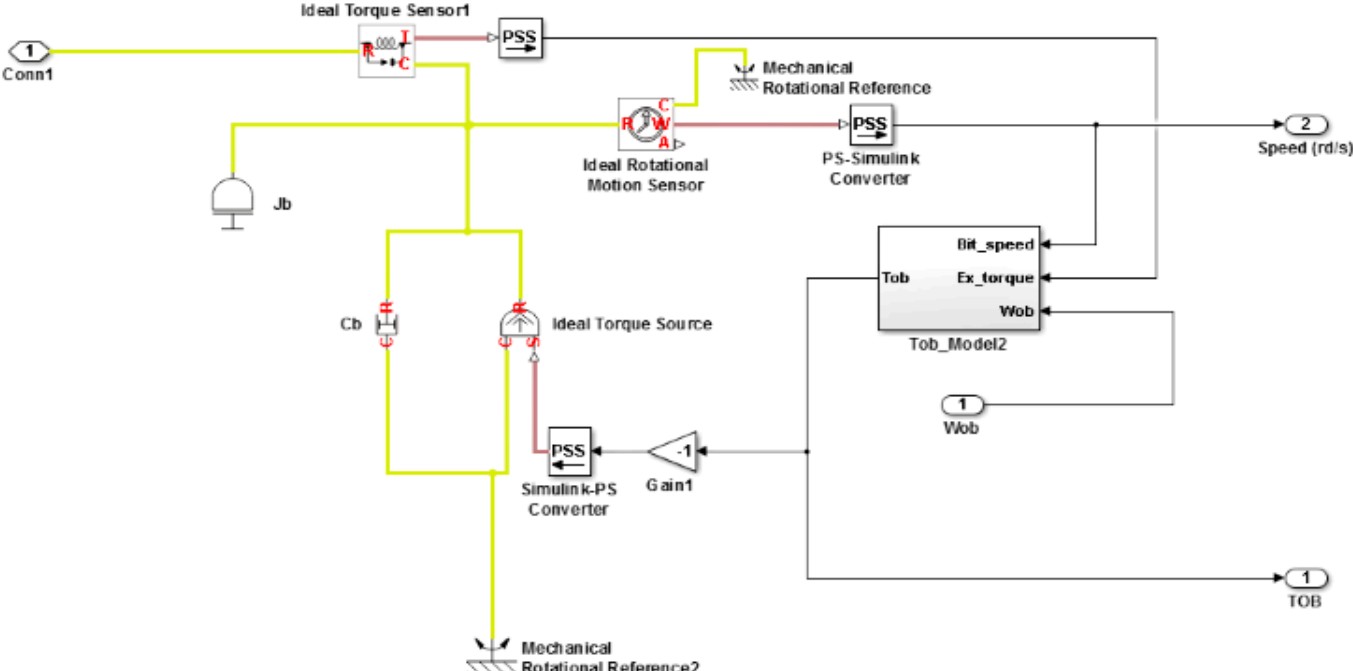

**Figure 2.** Simscape/Matlab Block diagram of Rock-bit interaction simulation model.

### 2.3.1. Scenario 1: Constant Weight on Bit

In this simulation scenario, the drilling system has been driven under constant Wob (*154 kN ≈ 16 t*) while varying the supply voltage of the top drive motor [42]. The obtained Tob and the supply voltage are shown in Figure 4a,b, respectively, the velocities of top drive and the BHA are illustrated in Figure 5a, and the stick-slip severity is given in Figure 5b. It can be noticed that the vibrations were very high at the beginning (class 3), then from $t = 13$ s they become within the permissible vibration range (classes 0 and 1). Because of the staircase descent of the top drive supply voltage, the severe stick-slip vibrations have been minimized to remain at a secure severity level, which cannot be guaranteed if we start with a supply voltage of 250 v. A similar scenario has been run with the staircase ascent of

the voltage as shown in Figure 6a. The drilling system is under severe state of stick-slip vibrations (class 3) with severity greater than 200%, as shown in Figure 7a,b. Figure 6b shows the resultant torque on bit for this scenario. To exit the high frequency stick-slip vibrations mode, the top drive must be operated with a practically high supply voltage, and then gradually brought back to its nominal voltage (250 Vdc), as highlighted in Figure 6a.

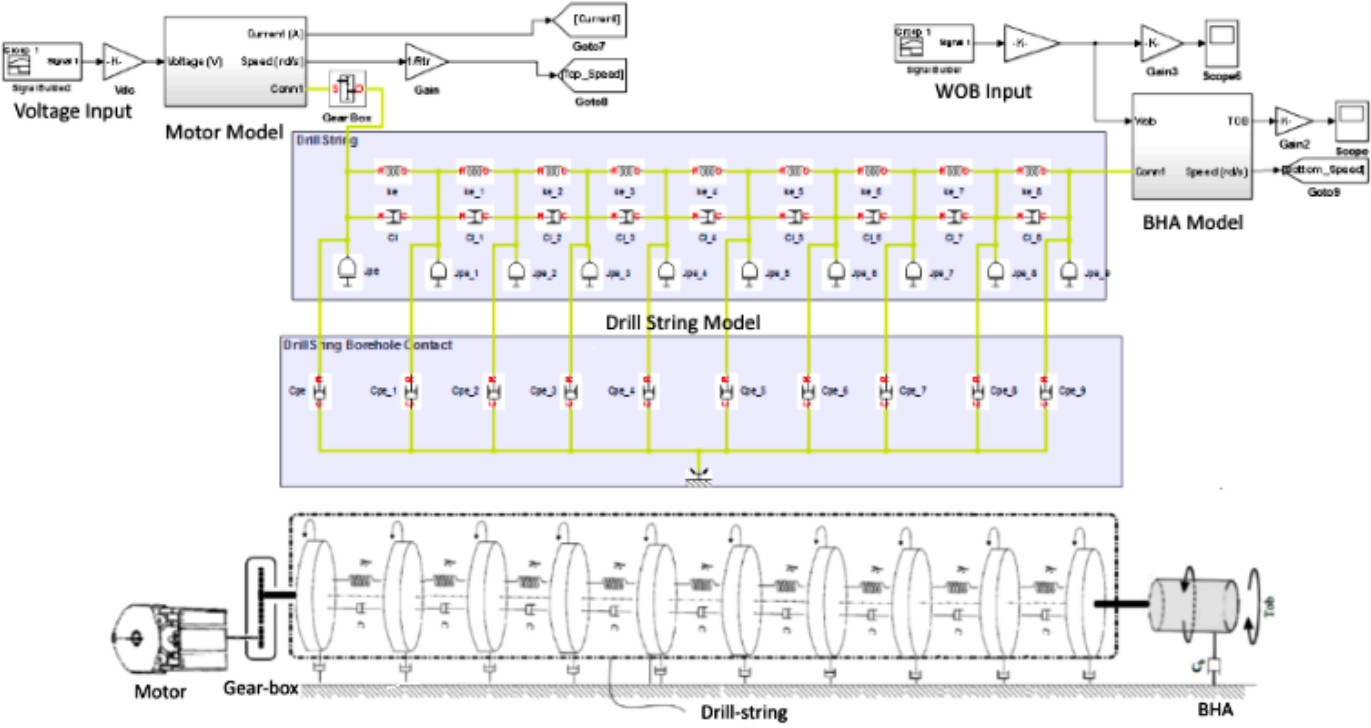

**Figure 3.** Rotary drilling system simulation model with ten degrees of freedom.

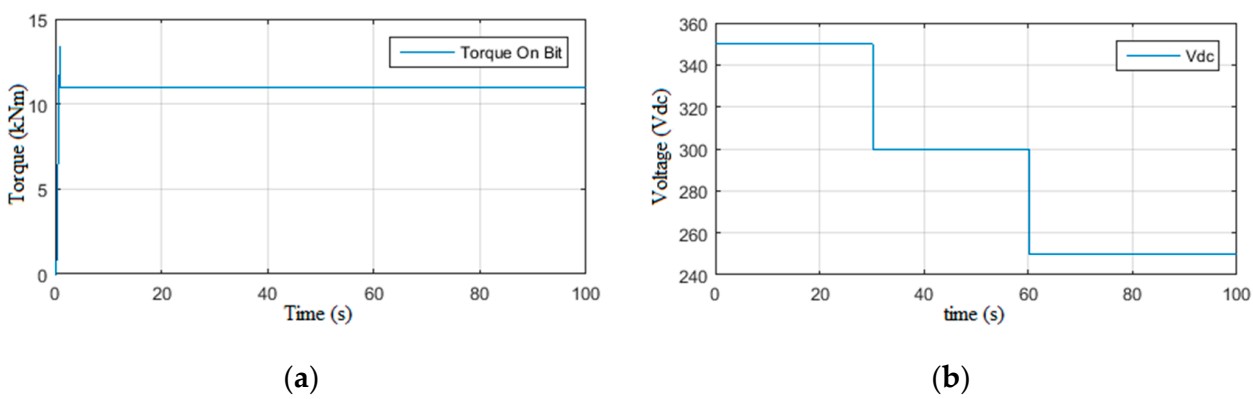

(**a**)　　　　　　　　　　　　　　　　(**b**)

**Figure 4.** Open loop Responses of Top drive motor for Scenario 1.1: (**a**) Torque on bit, (**b**) Input voltage.

### 2.3.2. Scenario 2: Constant Power Supply Voltage

In this second scenario, the drilling system is driven under constant power supply voltage (200 Vdc) while varying the weight on bit [35]. At the beginning, the Wob is considered constant, and perturbation is applied at $t = 30$ s (Figure 8b). This directly affects the torque on bit, as shown in Figure 8b, which causes the occurrence of torsional vibrations, as shown in Figure 8c. Consequently, the system momentarily enters the class (3) severity level as illustrated in Figure 8d. These vibrations have been remarkably amplified by increasing the Wob at $t = 60$ s when the system enters class (3) severity level permanently.

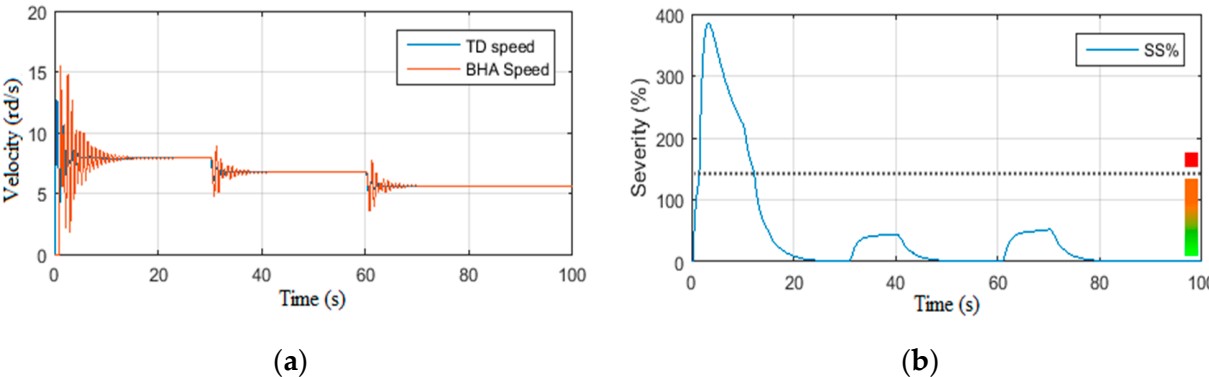

**Figure 5.** Scenario 1.1 open loop responses: (**a**) Rotational Velocities of Top drive and Drill bit, (**b**) Stick-slip severity.

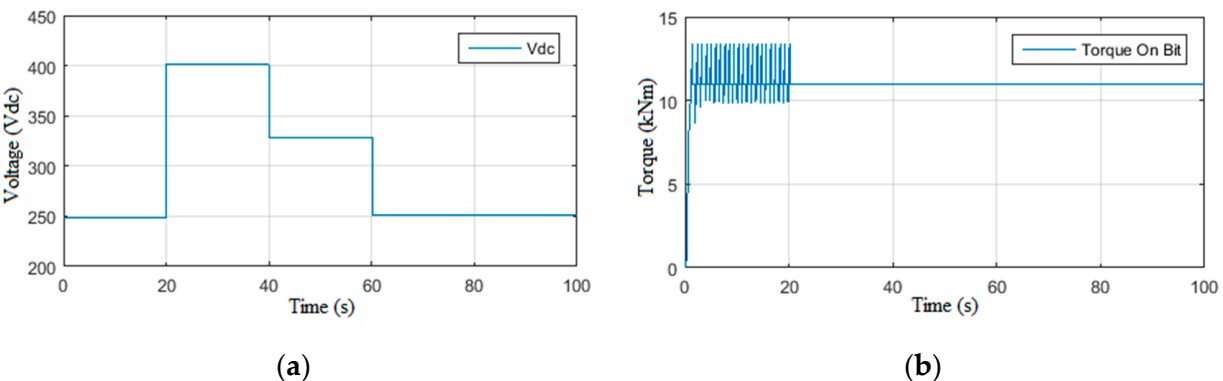

**Figure 6.** Responses of open loop scenario 1.2: (**a**) Voltage of the load, (**b**) Torque on bit (Tob).

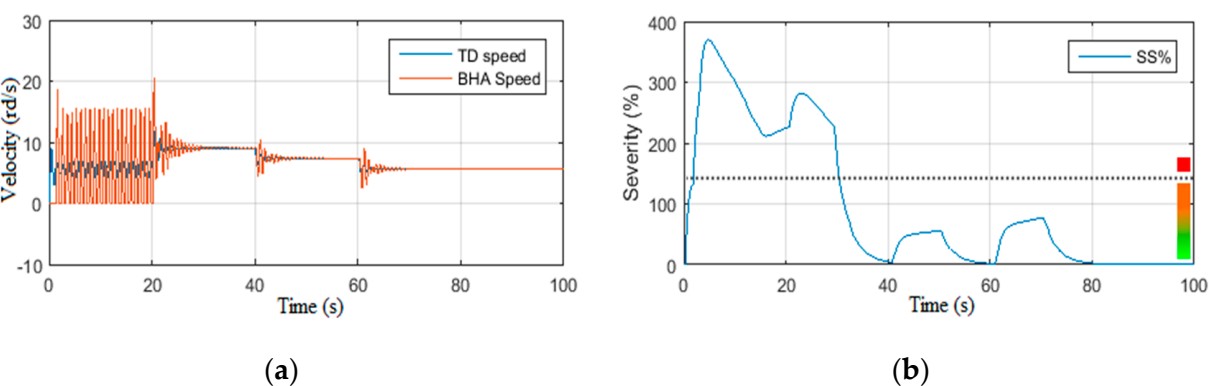

**Figure 7.** Scenario 1.2 open loop responses: (**a**) Rotational Velocities of Top drive and Drill bit, (**b**) Stick-slip severity.

In the scenario 2.2, the Wob is ramped up to 154kN, which is the threshold where stick-slip vibrations occurred in the scenario 2.1, as shown in Figure 9a. At this limit, the Wob is maintained constant. Thus, the drilling system is driven without stick-slip or even torsional vibration as given in Figure 9c,d. At *t* = 60 s, a disturbance takes place on the Wob, after which the drill-string is subjected to severe high frequency stick-slip vibrations (Figure 9c), and the system enters the class (3) severity in a permanent way, as demonstrated in Figure 9d.

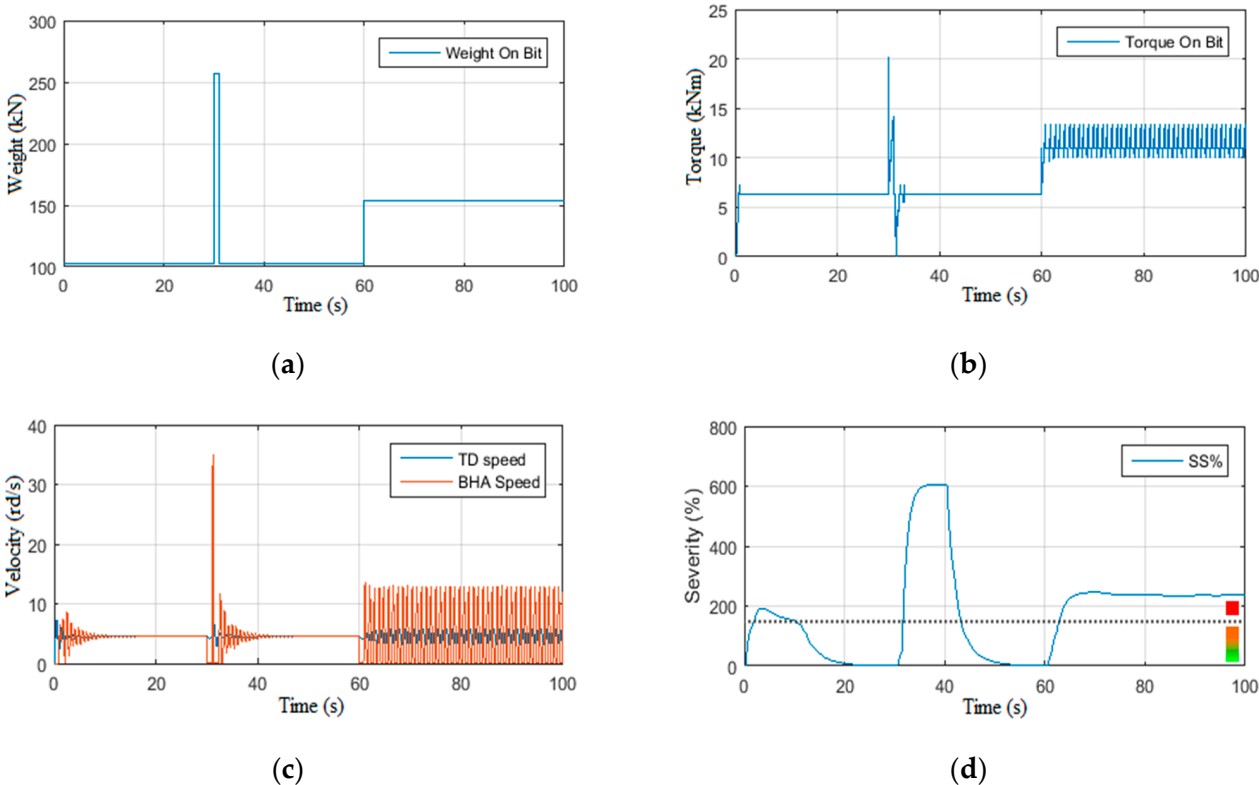

**Figure 8.** Scenario 2.1 responses: (**a**) Weight on bit, (**b**) Torque on bit, (**c**) Rotational velocities, (**d**) Stick-Slip severity.

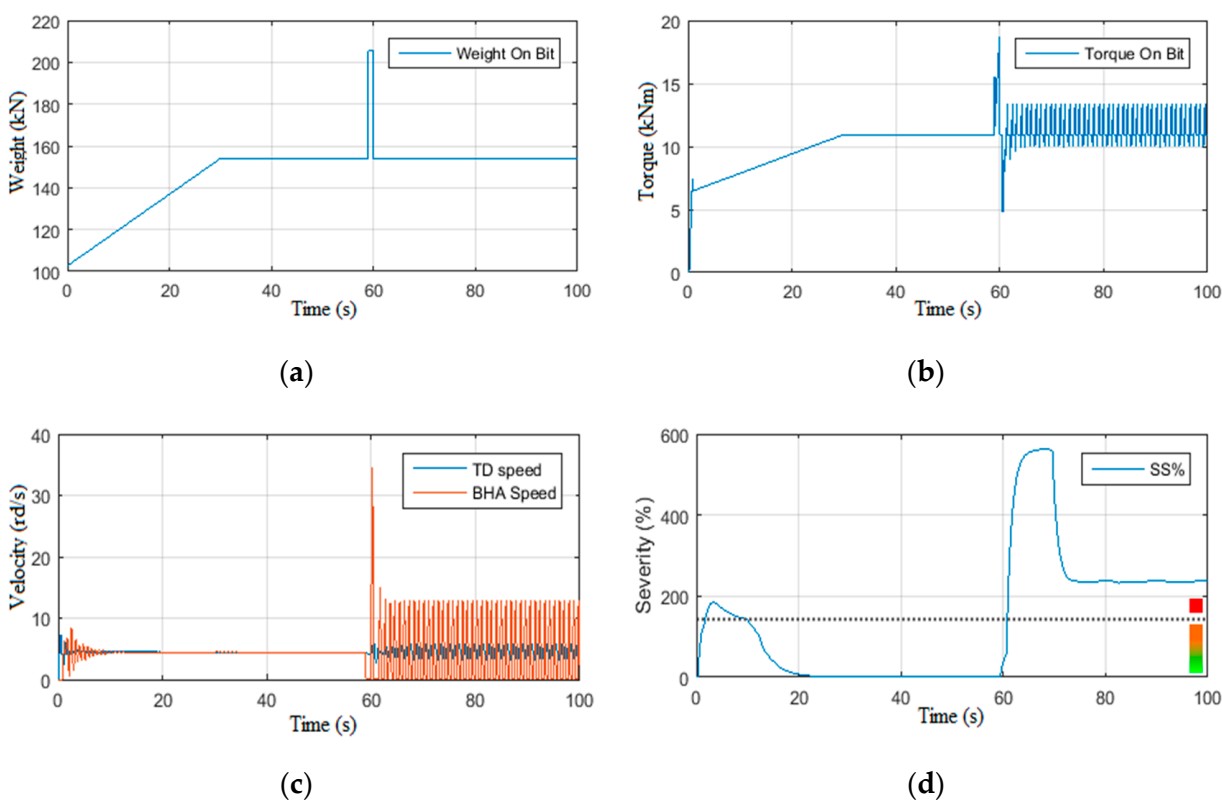

**Figure 9.** Responses of Scenario 2.2: (**a**) Weight on bit, (**b**) Torque on bit, (**c**) Rotational Velocities, (**d**) Stick-slip severity.

## 3. H∞ Observer-Based Controller Design

The H∞ observer-based controller designed in this study is a dynamic output feedback approach with two-stage cascaded structure as shown in Figure 10a,b [39,43]. In the first stage, the control system generates an estimate of drill bit velocity to be controlled using the measured speed and current of the top-drive outputs and the known input, which is the top-drive power supply voltage [42]. This estimation's generation is obtained by the designed observer with unknown Tob input. Then, the estimation is iteratively corrected till the estimated states are close enough to the measured states with an acceptable threshold error. Since in this study the constructed model has ten degrees of freedom, the resisting torque appears at the drill bit in the form of cutting and frictional torques [17]. Henceforth, the control strategy that will be considered is the drill bit velocity stabilization around a given nominal value under the presence of the resisting torque [25].

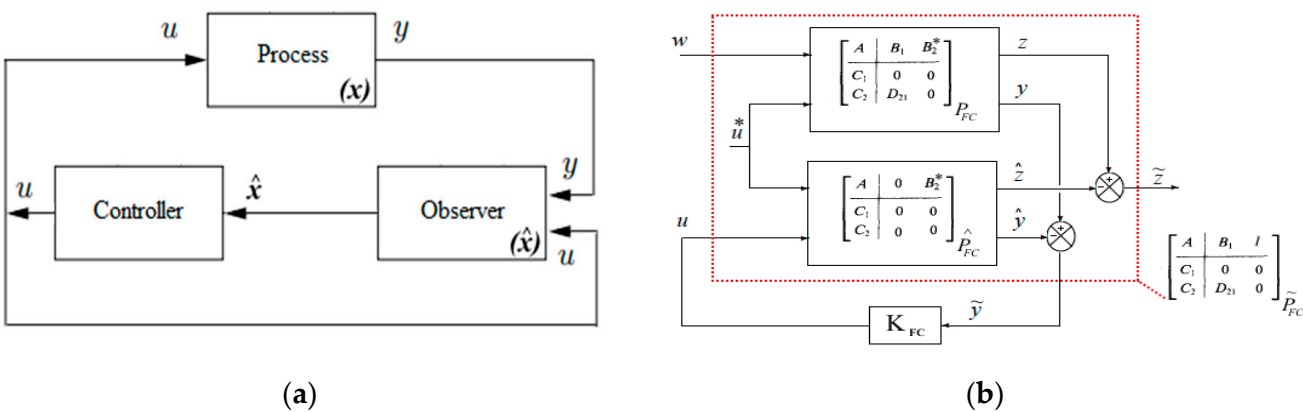

(**a**)                                                                                          (**b**)

**Figure 10.** Observer-Based controller for rotary drilling system: (**a**) General Scheme, (**b**) H∞ approach.

### 3.1. H∞ Approach

The development of an observer-based controller using the H∞ approach is reformulated to solving particular case of standard H∞ problem. Therefore, the mathematical synthesis of the state model (4) is mandatory for designing an observer with unknown input cascaded with the controller [39]. Figure 11 shows the closed loop form of drill-string model with the designed H∞ observer-based controller in the rotary drilling system.

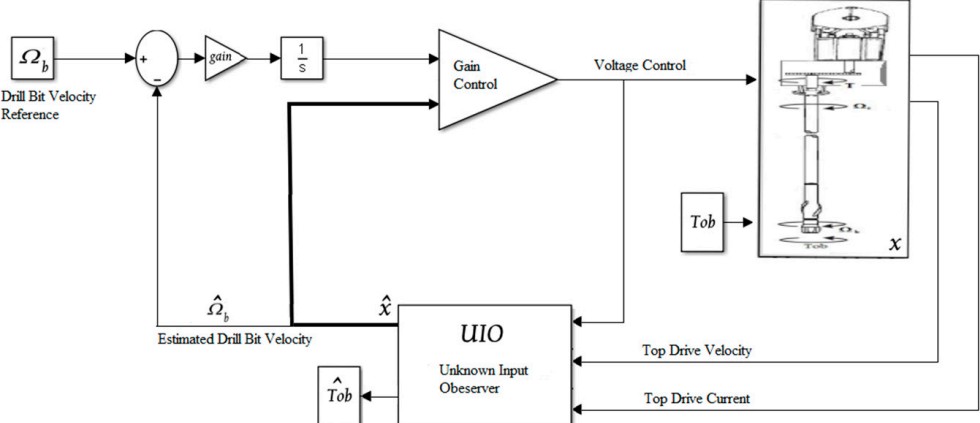

**Figure 11.** General Scheme of the rotary drilling system with Observer-based H∞ design.

3.1.1. Observer Synthesis with Unknown Inputs by H∞

Consider the augment system (5) obtained from the original state model (4). The system has been rewritten in this state space form in order to analytically verify its observability and controllability through solving the Riccati equations. In addition to that, this form can be seen as a full H∞ control problem, from which the gain of the H∞ observer-based controller will be calculated [21,36].

$$
\begin{cases}
\begin{pmatrix} \dot{x}(t) \\ \dot{T}_{ob}(t) \end{pmatrix} = \begin{pmatrix} A & E \\ 0 & 0 \end{pmatrix} \begin{pmatrix} x(t) \\ T_{ob}(t) \end{pmatrix} + B_1 W(t) + \begin{pmatrix} B_2 \\ 0 \end{pmatrix} u(t) \\[3mm]
z(t) = C_{1g} \begin{pmatrix} x(t) \\ T_{ob}(t) \end{pmatrix} \\[3mm]
y(t) = \begin{pmatrix} c_2 & 0 \end{pmatrix} \begin{pmatrix} x(t) \\ T_{ob}(t) \end{pmatrix} + D_{21} W(t)
\end{cases}
\tag{5}
$$

In model (5), the matrices $B_1$, $C_{1g}$, $D_{21}$ are chosen in an appropriate way to fulfill the necessary and sufficient conditions of the H∞ approach. Moreover, the matrices $B_i$ represent an unknown input observer designed by the tuning procedure. Therefore, we take $B_1 = \begin{pmatrix} b_1 & 0_{5\times2} \end{pmatrix}$, $D_{21} = \begin{pmatrix} 0_{2\times5} & I_{2\times2} \end{pmatrix}$, and

$$
C_{1g} = \begin{pmatrix}
1e^{-6} & 0 & 0 & 0 & 0 \\
0 & 1e^{-6} & 0 & 0 & 0 \\
0 & 0 & 1e^{-6} & 0 & 0 \\
0 & 0 & 0 & 5 & 0 \\
0 & 0 & 0 & 0 & 0
\end{pmatrix}
\text{ with } b_1 = \begin{pmatrix}
1 & 0 & 0 & 0 & 0 \\
0 & 1 & 0 & 0 & 0 \\
0 & 0 & 1 & 0 & 0 \\
0 & 0 & 0 & 1e^2 & 0 \\
0 & 0 & 0 & 0 & 27e^3
\end{pmatrix}
\tag{6}
$$

The vector $W(t)$ constitutes the perturbations on the states of the model and the measurements. It should be noted that there is no requirement on disturbances for the H∞ approach unlike the *LQG* approach for instance [24,44]. Therefore, the observer ensuring the performance criterion is given by (7).

$$
\begin{cases}
\dot{\hat{x}}_z(t) = \left( A_g - L_h C_z \right) \hat{x}_z(t) + B_z u(t) + L_h y(t) \\
\hat{z}(t) = C_{1g} \hat{x}_z(t) \\
\hat{y}(t) = C_z \hat{x}_z(t)
\end{cases}
\tag{7}
$$

with $A_g = A_z = \begin{pmatrix} A & E \\ 0 & 0 \end{pmatrix}$ and $L_h$ is the gain observer such that

$$
L_h = -Y_\infty C_z^T
\tag{8}
$$

$Y_\infty$ is the solution of the Riccati Equation (9), which guarantees the observability of the expanded system [36].

$$
A_g Y + Y A_g^T + Y \left( \gamma^{-2} C_{1g}^T C_{1g} - C_Z^T C_z \right) Y + B_z B_z^T = 0
\tag{9}
$$

3.1.2. Controller Synthesis by H∞

The drill bit velocity controller has been developed based on a reference model [45], i.e., the velocity of the drill bit is forced to follow the proposed reference velocity model. Thus, the model given by (7) is augmented to the model described by (10).

$$\begin{cases} \begin{pmatrix} \dot{\rho}(t) \\ \dot{x}(t) \\ \dot{\zeta}(t) \end{pmatrix} = \begin{pmatrix} \alpha_m & 0 & 0 \\ 0 & A & 0 \\ -\alpha c_m & \alpha c_r & 0 \end{pmatrix} \begin{pmatrix} \rho(t) \\ x(t) \\ \zeta(t) \end{pmatrix} + \begin{pmatrix} b_m & 0 \\ 0 & b_1 \\ 0 & 0 \end{pmatrix} \begin{pmatrix} r(t) \\ w(t) \end{pmatrix} + \begin{pmatrix} 0 \\ B_2 \\ 0 \end{pmatrix} u(t) \\ \\ z(t) = C_m \begin{pmatrix} \rho(t) \\ x(t) \\ \zeta(t) \end{pmatrix} + D_{12} u(t) \\ \\ y(t) = \begin{pmatrix} \rho(t) \\ x(t) \\ \zeta(t) \end{pmatrix} \end{cases} \quad (10)$$

where $\alpha_m$, $b_m$, $C_m$ are the state matrix, control, and observer of the reference model respectively, with $\alpha_m = -0.7$, $b_m = 0.7$, $C_m = 1$, $r(t)$ is the input of the reference model, and $\rho(t)$ is the output of the reference model, which the drill bit velocity should follow. $\zeta(t)$ is the integral of tracking error, and $\alpha$ is the controller tuning parameter, with $\alpha = 10$. The matrices $b_1$, $D_{12}$ are chosen in an appropriate way. Moreover, the two matrices $C_{1m}$ and $D_{12}$ constitute the tuning matrices required to bring back the response of the designed controller to the desired performance, where:

$$b_1 = \begin{pmatrix} 1 & 0 & 0 & 0 \\ 0 & 1 & 0 & 0 \\ 0 & 0 & 1 & 0 \\ 0 & 0 & 0 & 1 \end{pmatrix}, C_{1m} = \begin{pmatrix} 0 & 0 & 0 & 10 & 0 & 5 \\ 0 & 0 & 0 & 0 & 0 & 0 \end{pmatrix} \text{ and } D_{12} = \begin{pmatrix} 0 \\ 1 \end{pmatrix} \quad (11)$$

In a more compact form, the system (11) is rewritten as:

$$\begin{cases} \dot{x}_m(t) = A_m x_m(t) + \beta_1 W(t) + \beta_2 u(t) \\ z(t) = C_{1m} x_m(t) + D_{12} u(t) \\ y(t) = x_m(t) \end{cases} \quad (12)$$

Then, the control ensuring internal stability [25] and guaranteeing the performance criterion is written as:

$$u(t) = F_h x_m(t) \quad (13)$$

with

$$F_h = -\beta_2^T X_\infty \quad (14)$$

where $X_\infty$ is the solution of the Riccati Equation (15), which guarantees the controllability of the system [36].

$$A_m^T X + X A_m + X_m \left( \gamma^{-2} \beta_1 \beta_1^T - \beta_2 \beta_2^T \right) X_m + C_{1m}^T C_{1m} = 0 \quad (15)$$

## 4. Results and Discussion

Many scenarios have been considered in this part in order to study and test the robustness of the designed H∞ observer-based controller for different situations in the drilling field. First, the estimation precision of the designed observer without any controller has been investigated, and then the controller has been activated in order to evaluate the complete H∞ observer-based controller bloc. The parameters used in these scenarios are namely the moments of inertia of the rotating parts, the stiffness of the drill pipe, and the viscous friction coefficients. These parameters have been calculated from the basic equations and the fundamental rig parameters, such as: the length of the drill pipes and the drill collars, the diameter and thickness of the drill-string and drill presses, the mud viscosity, etc.

### 4.1. Observer Performance Tests

In these tests, the controller has been disabled while applying the following scenarios to study the estimation precision of the designed observer.

#### 4.1.1. Scenario 3: Constant Wob with Step Voltage

A constant Wob equal to 0.7 of the overall drilling weights, which is equivalent to 29 tons, has been applied over the simulation time. A step voltage that rises from 300 V to 450 V at $t = 10$ s has been also considered as shown in Figure 12a,b. Figure 13a shows the measured and the estimated drill bit velocities with the estimation errors. Hence, it can be noticed that the system suffers from high frequency stick-slip vibrations of class (3) as demonstrated in Figure 13b. Then, by increasing the top drive power supply voltage, the vibrations have become of class (1) at $t = 23$ s, which is a tolerable region. This operation takes a considerable time (13s) to eliminate the vibrations. Due to their high frequency content, such vibrations can create too much damage during such a time period [45,46]. In addition to that, the observer has provided good estimation of the unknown Tob, which is treated as a disturbance on the drill-string dynamic model, as given by Figure 14. It is also noticeable that the measurable states of the system, namely the Top drive velocity and current, have been estimated with good accuracy, as demonstrated in Figure 15a,b, respectively.

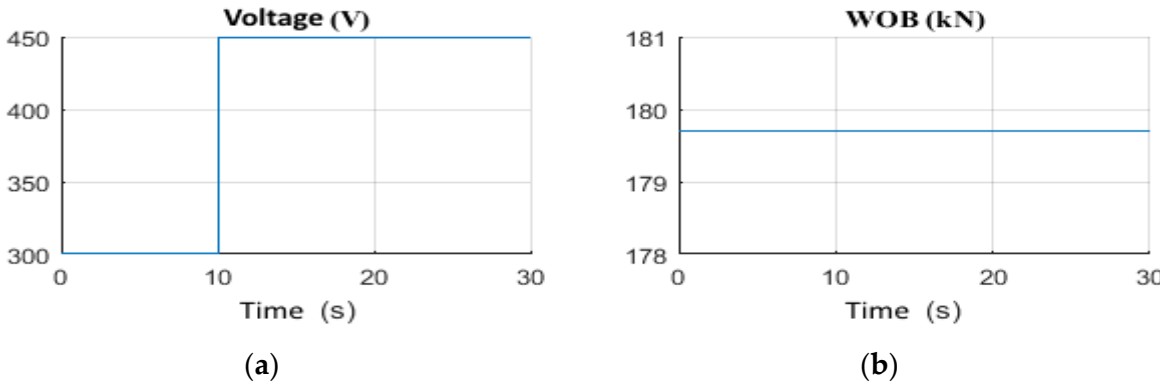

**Figure 12.** Test inputs of the observer for Scenario-3: (**a**) Top drive motor voltage, (**b**) Applied Weight on bit.

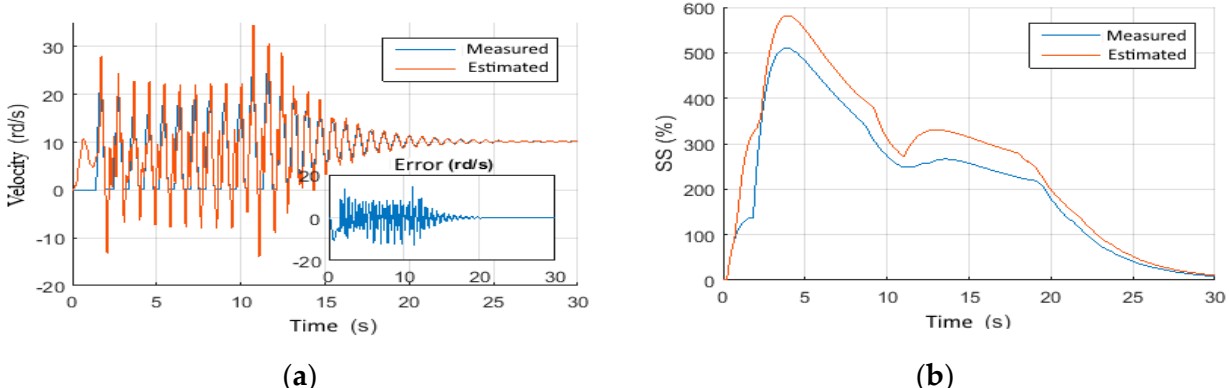

**Figure 13.** Measured and estimated variables by the designed observer: (**a**) Rotational velocities, (**b**) Stick-slip severity.

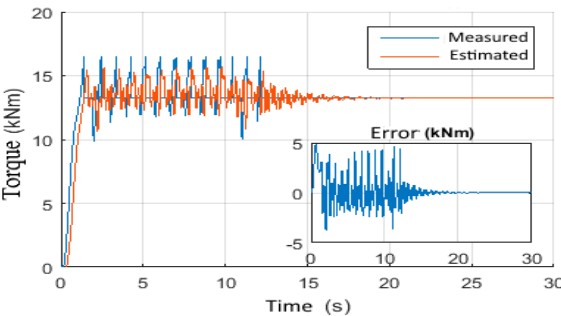

**Figure 14.** Measured and estimated Toque on bit by H∞ observer.

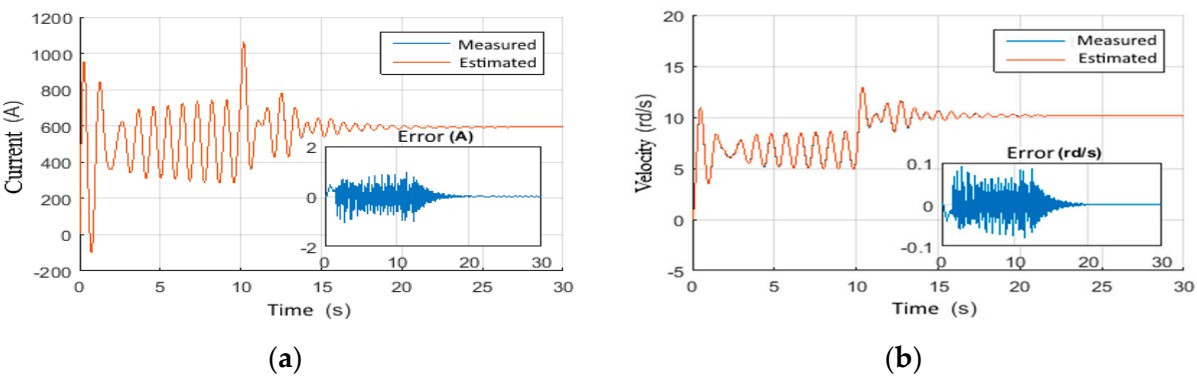

(**a**)　　　　　　　　　　　　　　　　　　(**b**)

**Figure 15.** Measured and estimated variables by the designed observer: (**a**) Top-drive current, (**b**) Top-drive velocity.

4.1.2. Scenario 4: Random Wob with Constant Voltage

Through this scenario, the observer behavior against random Wob variations has been examined. The randomness of this input is stimulated by the dynamic axis of the drill-string [47]. For this, the inputs shown in Figure 16a,b have been applied. The H∞ observer has provided good estimation results for the drill bit velocity as shown in Figure 17a, and for the unknown Tob input, as shown in Figure 17b. The measured inputs shown in Figure 18a,b of the top drive have a fluctuating behavior around 500A (for the current), and around 15 rd/s (for the velocity). The large peak of the current at the beginning is mainly due to the direct start of the motor without activation of the controller. Further, the velocity and the current fluctuations have been well estimated with acceptable accuracy, even with the presence of unknown Wob characterized by arbitrary dynamics, as shown in Figure 18a,b.

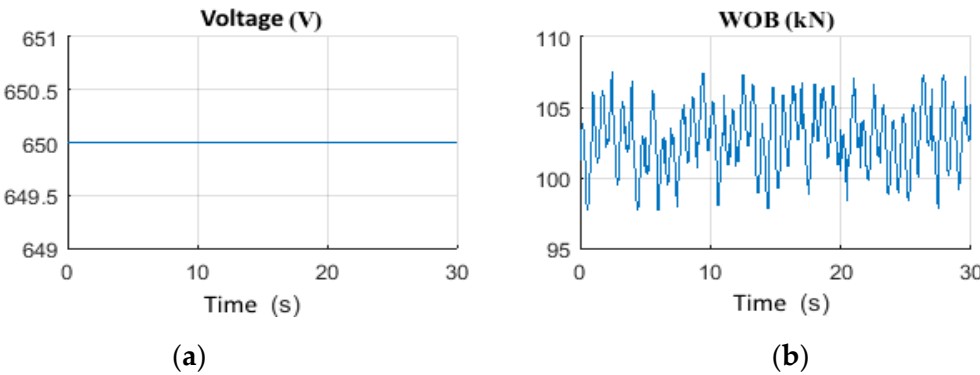

(**a**)　　　　　　　　　　　　　　　　　　(**b**)

**Figure 16.** Test inputs of the observer for Scenario-4: (**a**) Top drive motor voltage, (**b**) Applied Weight on bit.

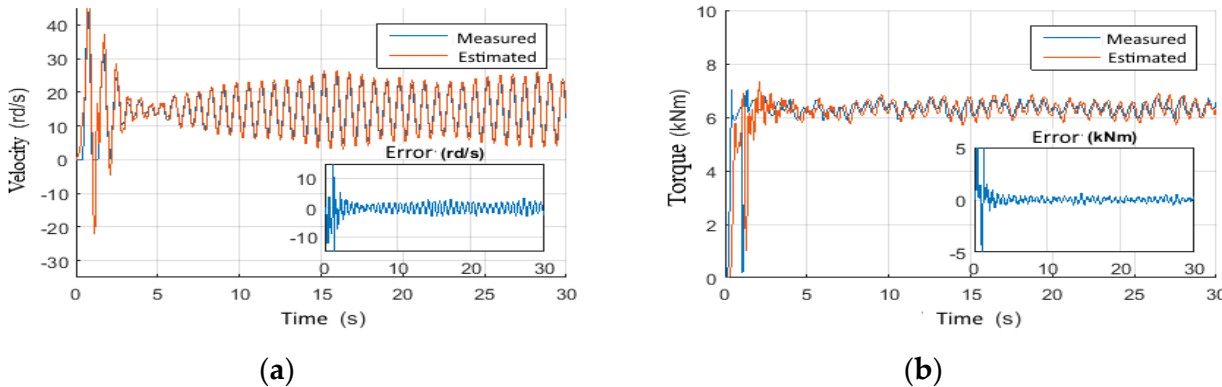

**Figure 17.** Measured and estimated variables by the designed observer for Scenario-4: (**a**) Rotational velocities, (**b**) Stick-slip severity.

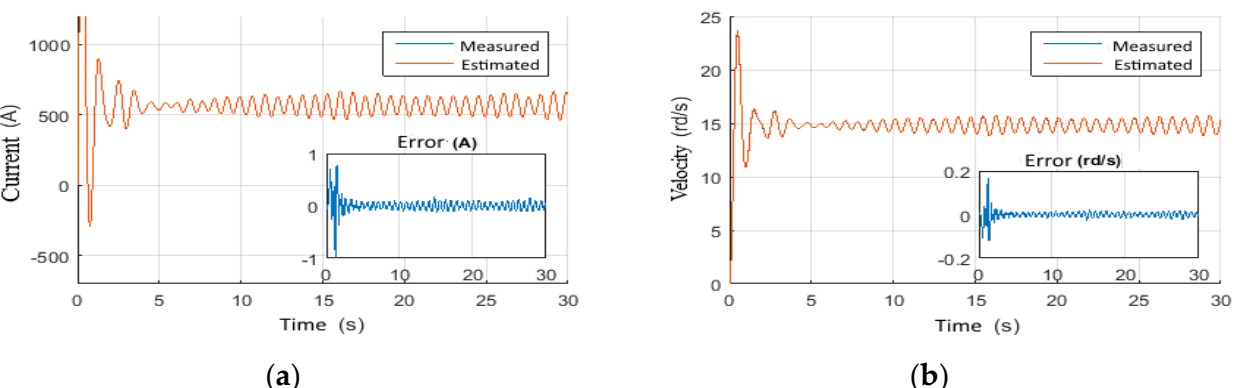

**Figure 18.** Measured and estimated variables by the designed Observer for Scenario-4: (**a**) Top-drive current, (**b**) Top-drive velocity.

*4.2. Scenario5: Disturbed Measurements*

In this scenario, the system has been driven with a constant voltage equal to 650 V and constant Wob equal to 103 kN, as demonstrated in Figure 19a,b. Then, disturbances have been introduced to the top drive current and velocity with non-zero mean signal with frequency of 1kHz as given in Figure 20a,b [48]. As expected, the H∞ observer has arrived to filter the unstructured perturbations unlike other observer, such as the Kalman filter. This is mainly caused by the supposition put on the choice of covariance matrices, whilst H∞ design did not require any restrictions in terms of their choices. Figure 21a,b show the estimated velocity and torque of the drill bit, and Figure 22a,b demonstrates the estimated current and velocity of the top drive respectively. It can be concluded that H∞ estimation performance has not been affected by the unstructured perturbations that can appear on the top drive inputs during the drilling process. Hence, the controller robustness can be maintained even in such situations, since the observer estimations results can have a direct influence on the controller robustness.

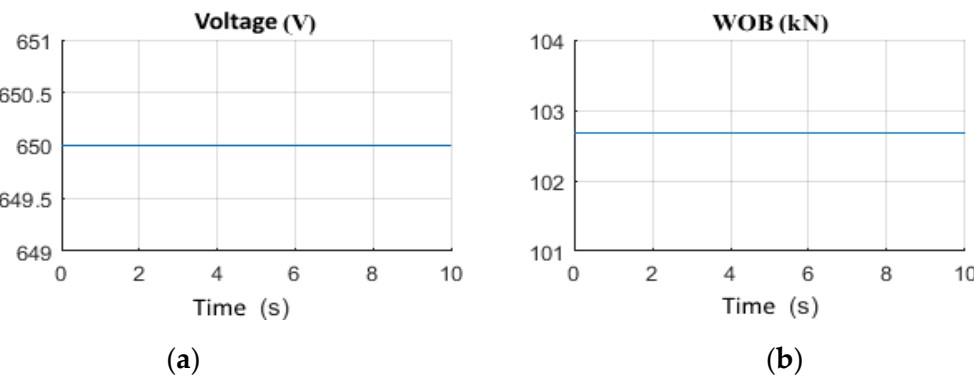

**Figure 19.** Test inputs of the observer for Scenario-5: (**a**) Top drive motor voltage, (**b**) Applied Weight on bit.

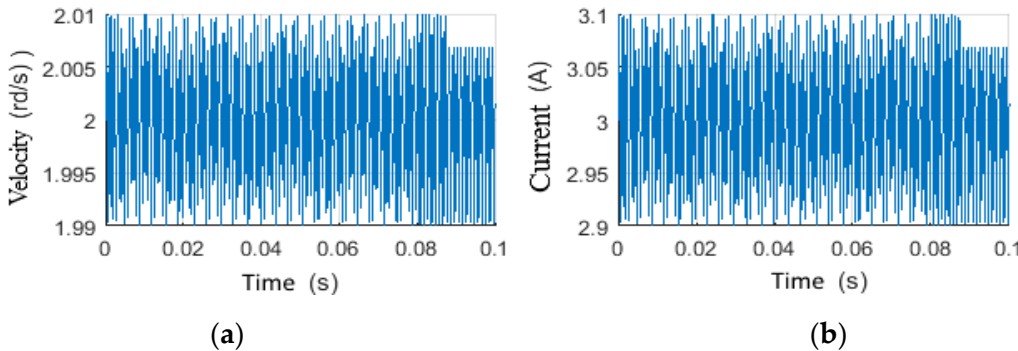

**Figure 20.** Noises on Top drive measurements, (**a**) Rotational velocity, (**b**) The current.

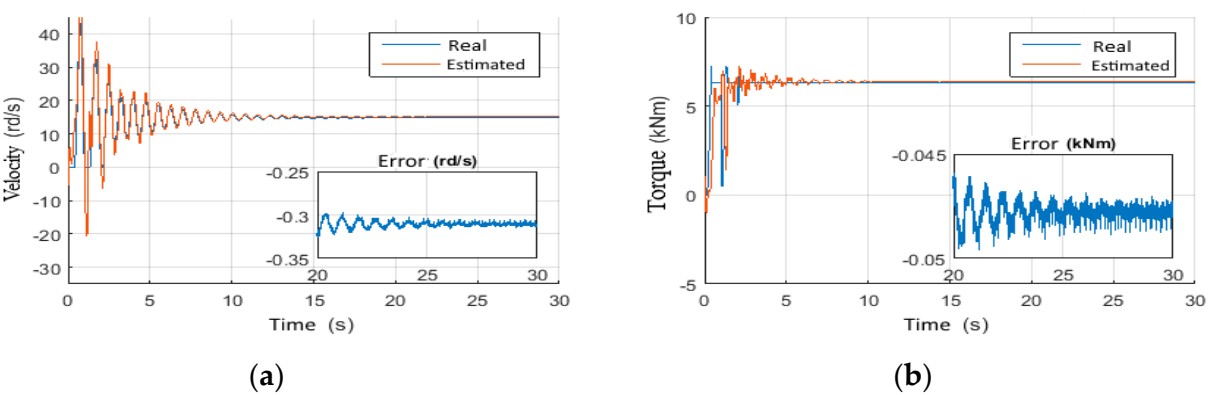

**Figure 21.** Measured and estimated variables by the designed Observer for Scenario-5: (**a**) Rotational velocities, (**b**) Tob.

### 4.3. Scenario6: Parametric Variation

This time the system's inputs are as follows: a constant supply voltage of 650 V with a constant Wob of about 103 kN as shown in Figure 23a,b [49]. However, a variation of the parameters, namely resistance (*r*), inductance (*l*), length, and mass of string, has been applied. These are related to the calculation of the moments of inertia and the viscous friction coefficients [50]. As shown in Figures 24 and 25, the designed observer has demonstrated good performance against the parametric variations. However, it showed sensitivity to the torque constant variation (*K*), as given in Figure 24b. The torque constant has a deviation of 3% from its nominal value [49].

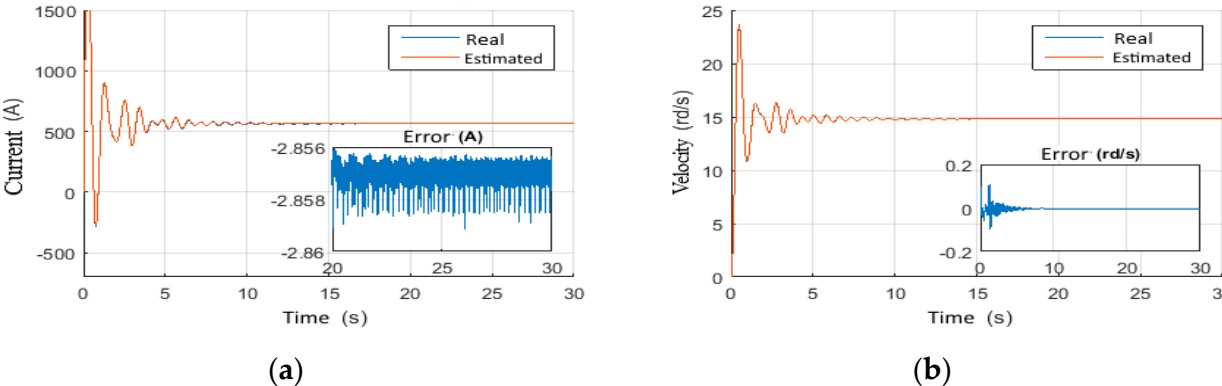

**Figure 22.** Measured and estimated variables by the designed Observer for Scenario-5: (**a**) Top-drive current, (**b**) Velocity.

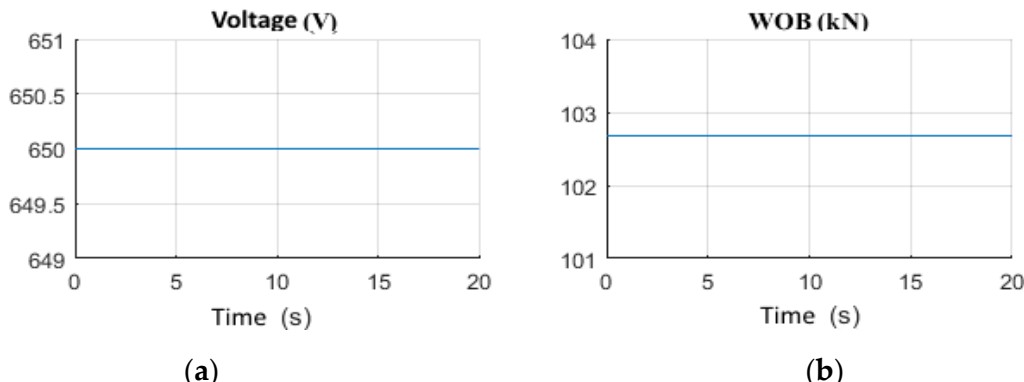

**Figure 23.** Test inputs of the observer for Scenario-6: (**a**) Top drive motor voltage, (**b**) Applied Weight on bit.

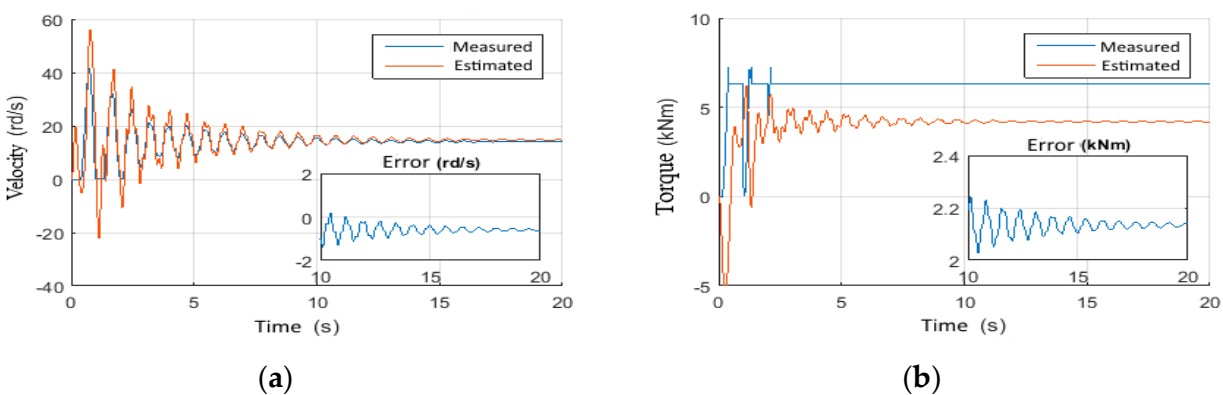

**Figure 24.** Measured and estimated variables by the designed Observer for Scenario-6: (**a**) Rotational velocities, (**b**) Tob.

*4.4. Performance Tests of Observer-Based Controllers*

In this part, the performances of the full H∞ observer-based controller have been discussed. Keeping in mind that the controller receives the estimated state in order to provide output feedback to the top drive [43], the main objective of the controller is to impose a predefined dynamic on the top drive that ensures the tracking of the desired reference by the drill bit [51].

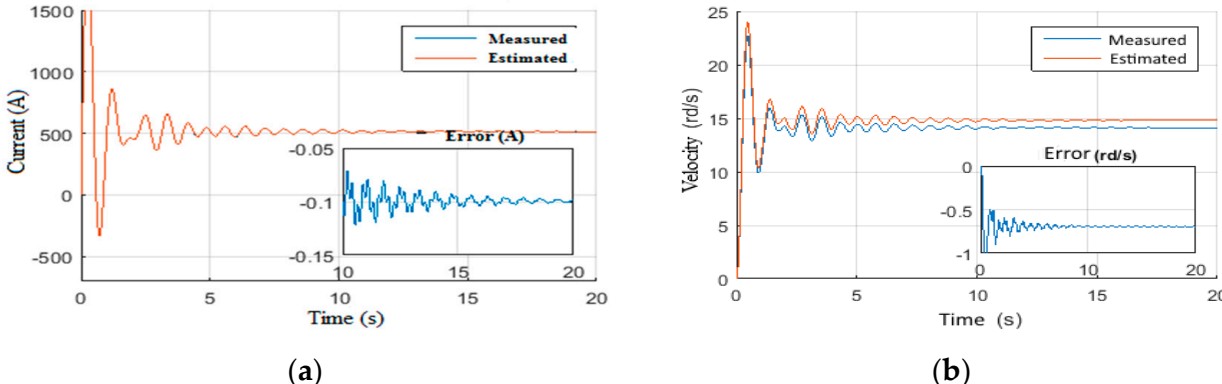

**Figure 25.** Measured and estimated variables by the designed Observer for Scenario-6: (**a**) Top-drive current, (**b**) Velocity.

### 4.4.1. Scenario 7: Voltage Step Reference Tracking

In this scenario, the control system is subjected to a step reference that rises from 0 to 7 rd/s, then from 7 rd/s to 15 rd/s without applying any additional weight on the bit. Figure 26a,b shows the velocity and voltage of top drive responses to this first input scenario. It is clear that the controller has guaranteed the tracking of the reference with acceptable error and within a short time. Figure 27a,b shows the controller output for the top drive current and velocity respectively. This demonstrates that the controller has provided smooth values with fewer fluctuations. Thus, this signal can be handled by the top drive without any abnormalities. Moreover, the increase of current consumption (at $t$ = 15 s) is caused by the increase in viscous friction due to the increased speed of the drill-string, which means that H∞ controller can be straightforwardly implementable in the top drive machines.

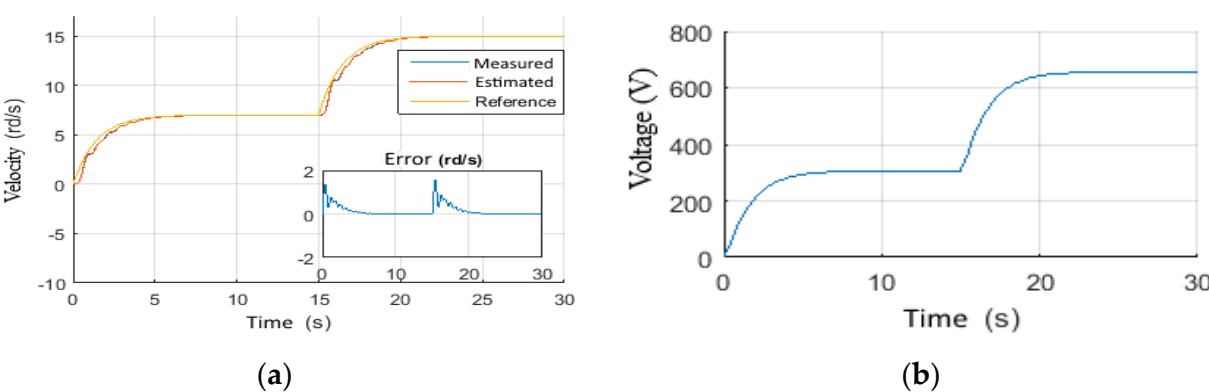

**Figure 26.** Observer-based controller responses: (**a**) Controlled drill bit velocity, (**b**) Control voltage.

### 4.4.2. Scenario 8: Wob step Tracking Reference

This time, the control system is subjected to a Wob step reference in order to test its robustness. To do this, the Wob has been increased from 25 kN to 180 kN at $t$ = 10 s (Figure 28a) with a reference velocity of 7 rd/s. It can be noticed from Figure 29a that this increase caused a complete stop of the drill bit for 2 s, followed by high frequency stick-slip vibrations (Figure 28b). Then, the designed controller forced the drill-string to leave this stuck phase and mitigate the vibrations by tracking the desired outputs in 7 s (Figure 29). Moreover, the control signals sent to the top drive are shown in Figure 30a,b.

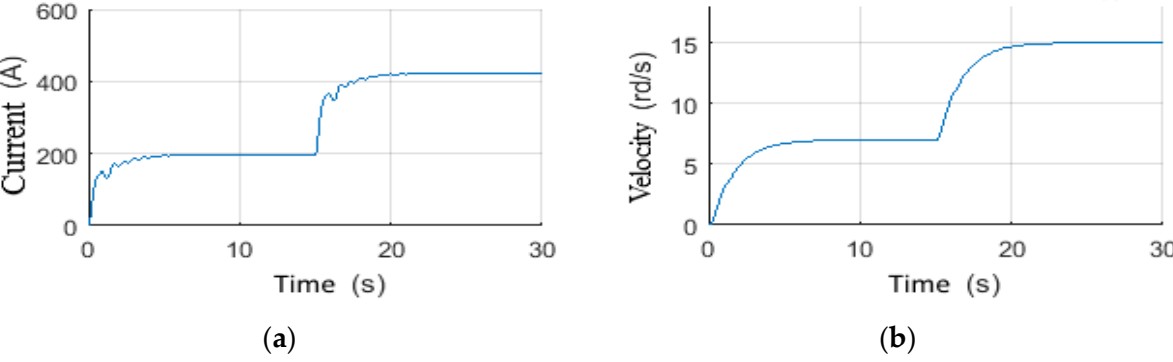

**Figure 27.** Observer-based controller responses for Top Drive: (**a**) Current, (**b**) Velocity.

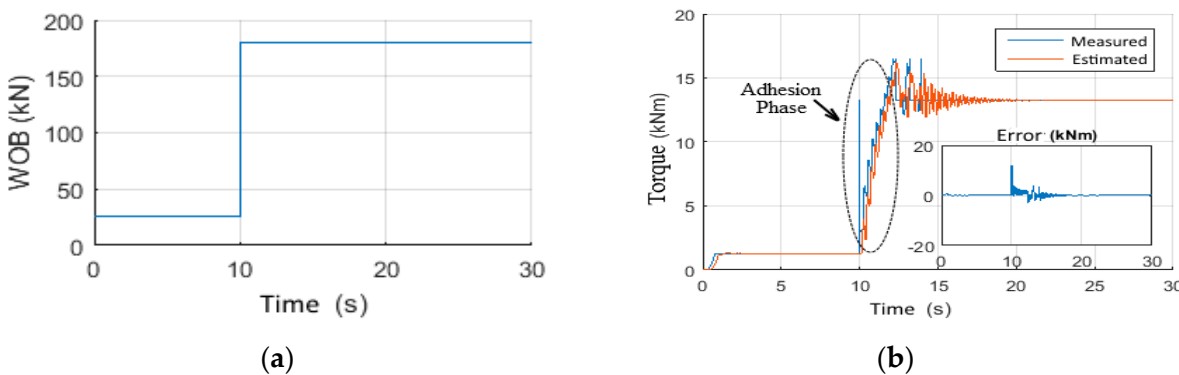

**Figure 28.** Observer-based controller responses for Scenario-8: (**a**) Wob, (**b**) Torque on bit.

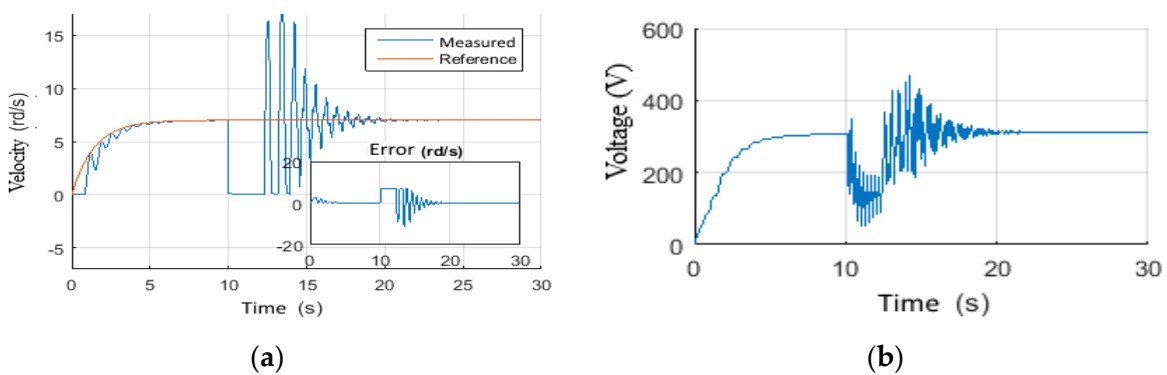

**Figure 29.** Observer-based controller responses for Scenario-8: (**a**) Velocity, (**b**) Voltage.

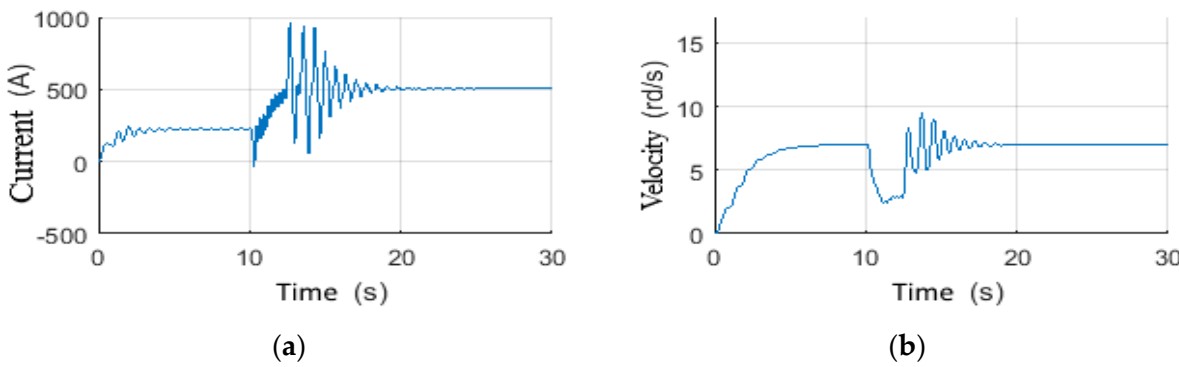

**Figure 30.** Observer-based controller responses of Top drive for Scenario-8: (**a**) Current, (**b**) Velocity.

### 4.4.3. Scenario 9: Stick-Slip Mitigation

In this scenario, the controller performance's improvement in suppressing the high frequency stick-slip vibrations has been investigated. The drilling system has been driven with a deactivated controller at the beginning and with *Wob* = 180 kN and a voltage of 300 V. These inputs cause the generation of the stick-slip phenomenon in the drill-string [9]. At *t* = 10 s, the controller has been activated, as shown in Figures 31 and 32a,b. These figures demonstrated that after 5s from controller activation, the vibrations have been reduced from class (3) to class (1). Then, they have been completely mitigated in a very short time in comparison to manual manipulation procedures used nowadays in the drilling fields. Moreover, the control law shown in Figure 33a,b is very smooth and can be injected into an operational top drive without too many technical restrictions.

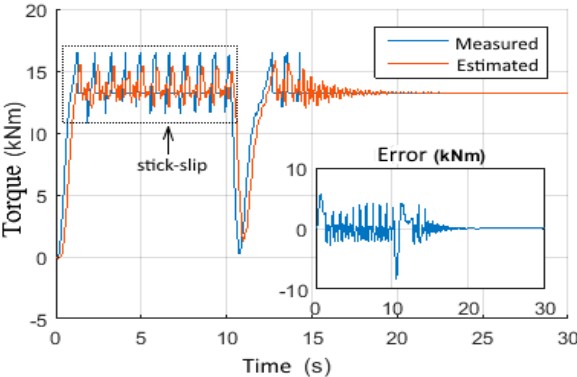

**Figure 31.** Controlled Torque on bit by H∞ approach.

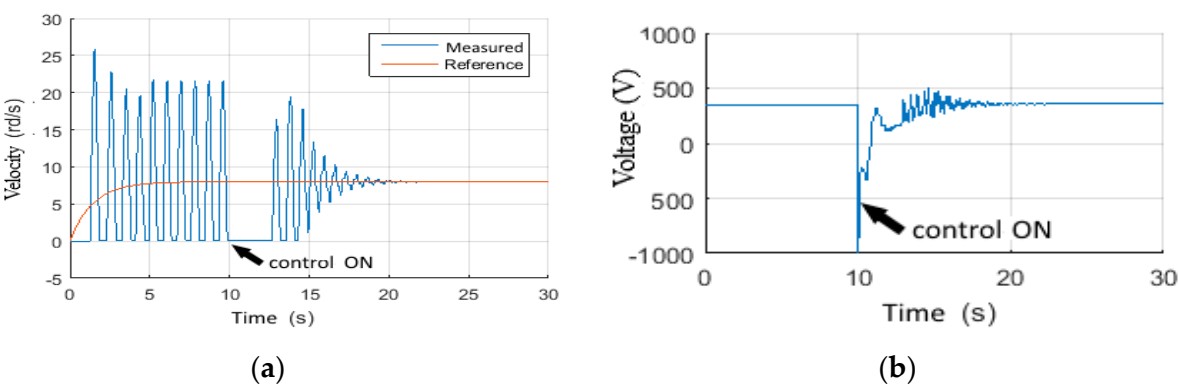

**(a)** **(b)**

**Figure 32.** Observer-based controller responses for Scenario-9: (**a**) Velocity, (**b**) Voltage.

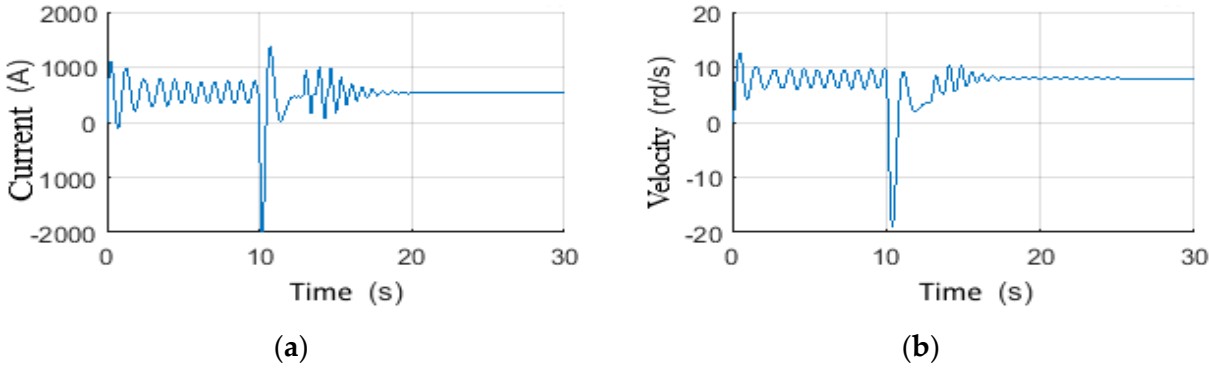

**(a)** **(b)**

**Figure 33.** Observer-based controller responses for Top drive, Scenario-9: (**a**) Current, (**b**) Velocity.

### 4.4.4. Scenario 10: Wob Disturbances Filtering

In this scenario, the ability of the designed approach to filter the unstructured disturbances on the Wob has been studied [24]. The perturbations are generally initiated by the axial dynamic of the drill-string, which has been previously neglected in the mathematical model of the drill-string [26]. First, the drilling system is driven with constant and disturbed Wob, and with deactivated control. Then, at $t = 60$ s, the controller has been activated. Figure 34 shows the attenuation of the velocity fluctuations of drill bit by the activated controller. The stick-slip severity has also been reduced from class (3) to class (1) by activating the controller, as shown in Figure 35a. Moreover, Figure 35b shows the required controller voltage that should be applied to the top drive. It can be noticed that this is a smooth and tolerable input.

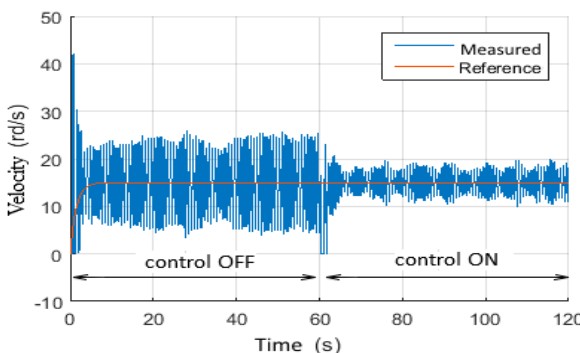

**Figure 34.** Noised drill bit velocity without and with H∞ control activation.

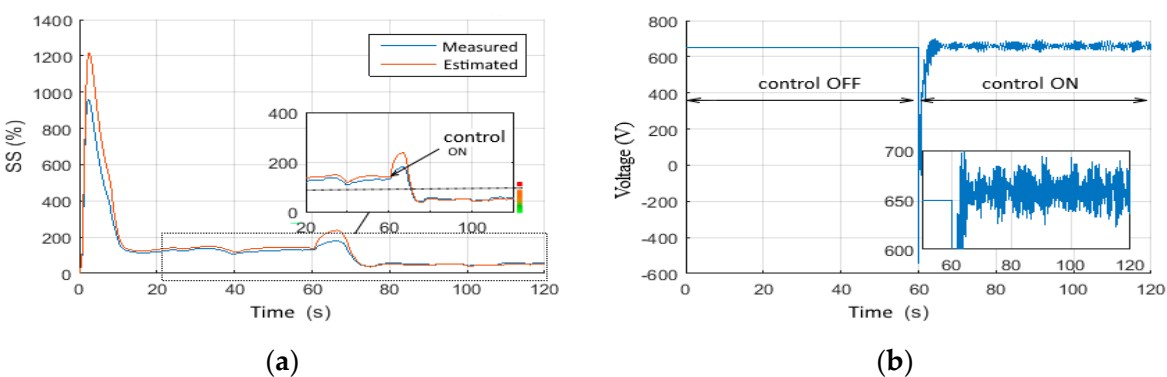

(a) (b)

**Figure 35.** Observer-based controller responses for Scenario-10: (**a**) Stick-Slip severity, (**b**) Voltage.

### 4.5. Scenario 11: Performance Limitations

In this scenario, the controllers' performances have been tested under extreme conditions by imposing disturbances on the Wob (Figure 36) and on both the top drive velocity and current (Figure 37a,b), along with parametric variations on the length of drill pipes and drill collars. These variations are of the order of 130% of the nominal value, plus a variation in the torque constant of the order of 103%. Figure 38 shows that the designed H∞ observer-based controller has provided a better estimation than LQG even for this scenario [44]. Moreover, the LQG control law shown in Figure 39a is very disturbed and cannot be implemented in operational top drive, while the H∞ has provided a better control law as shown in Figure 39b. Figures 40–42 show the obtained results of H∞ in comparison to LQG for torque on bit, top drive current, and top drive velocity, respectively. It is clear through these figures that the designed controller has provided better estimation and control than the LQG even for this scenario where extreme conditions have been considered. Therefore, we can conclude that the proposed H∞ observer-based controller is better than LQG and can guarantee better robustness in operating rotary drilling systems.

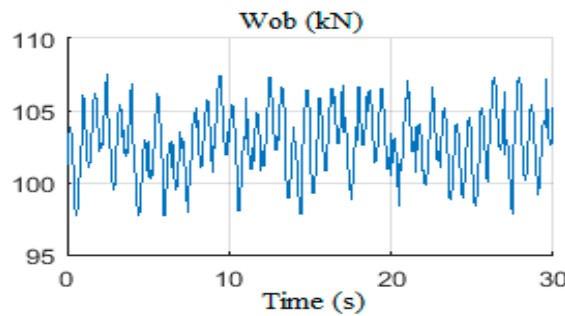

**Figure 36.** Perturbed Weight on bit input for Scenario-11.

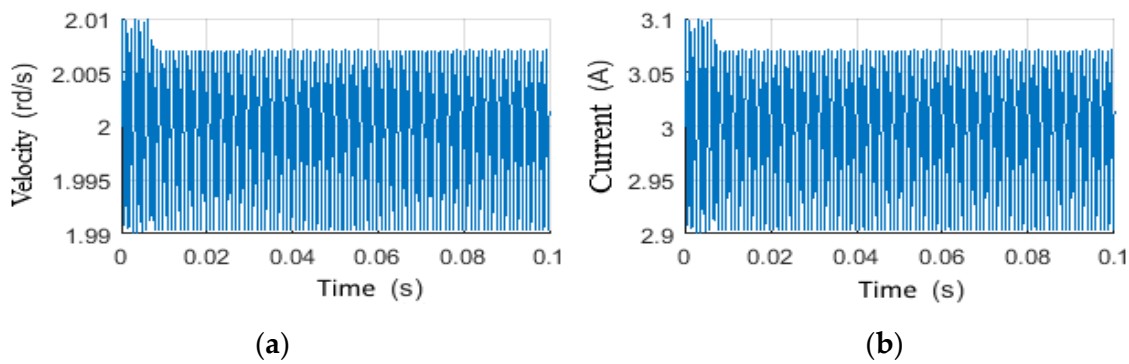

(**a**)                                                                                          (**b**)

**Figure 37.** Noised inputs of Top drive for Scenario-11: (**a**) Velocity, (**b**) Current.

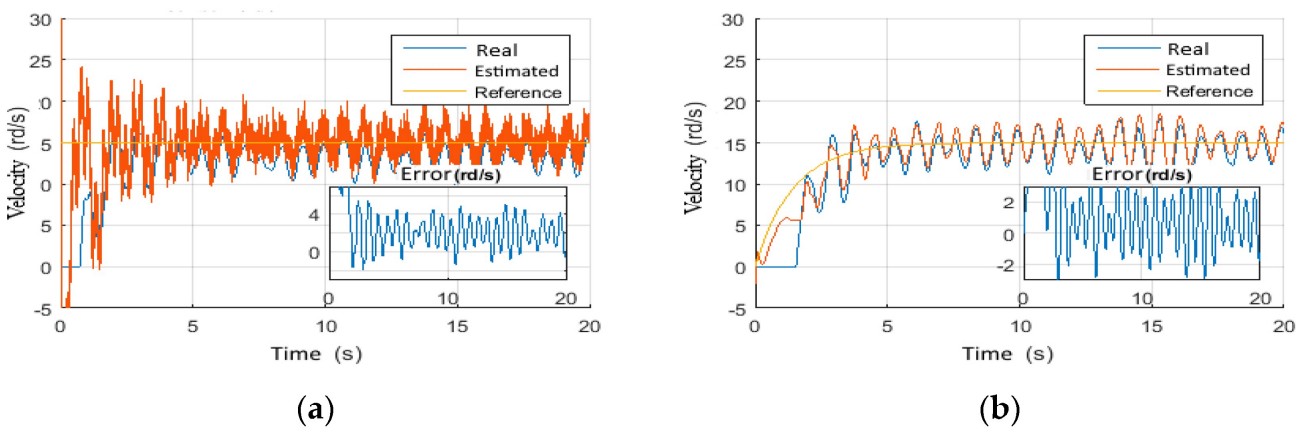

(**a**)                                                                                          (**b**)

**Figure 38.** Real and estimated Drill bit velocity with: (**a**) LQG controller, (**b**) H∞ controller.

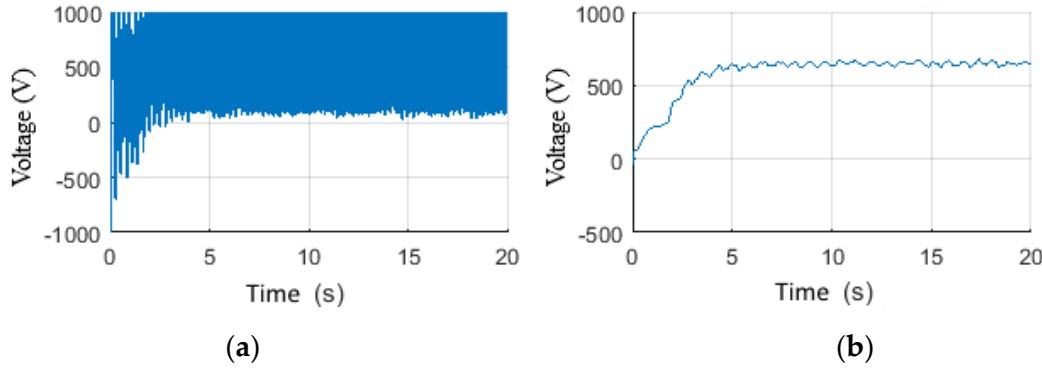

(**a**)                                                                                          (**b**)

**Figure 39.** Top drive voltage for Scenario-11 with: (**a**) LQG controller, (**b**) H∞ controller.

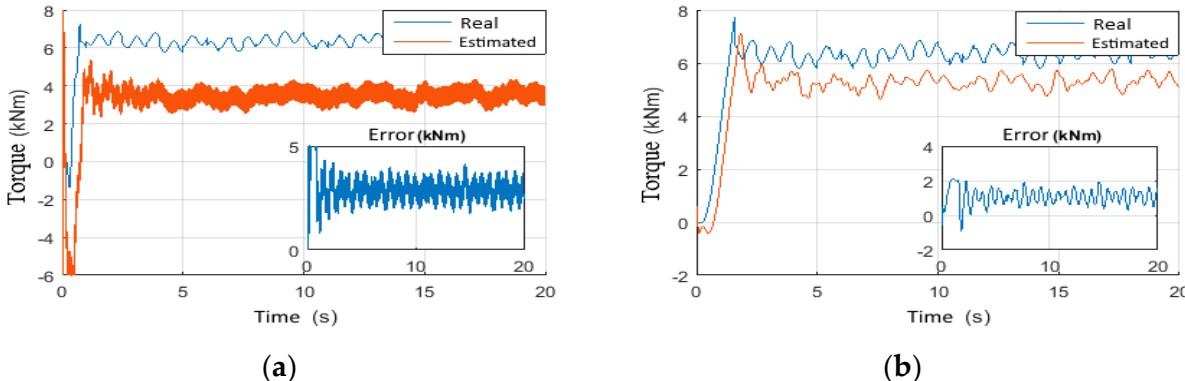

**Figure 40.** Real and estimated Torque on bit under: (**a**) LQG controller, (**b**) H∞ controller.

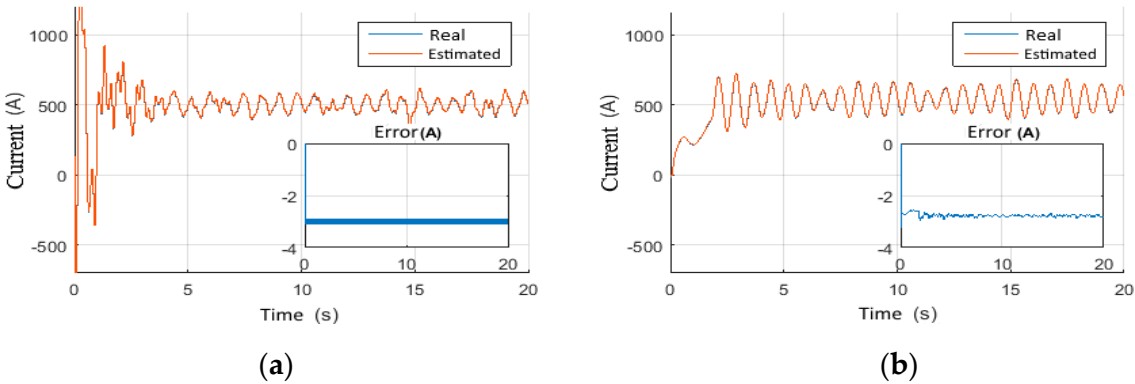

**Figure 41.** Real and estimated Top drive current under: (**a**) LQG controller, (**b**) H∞ controller.

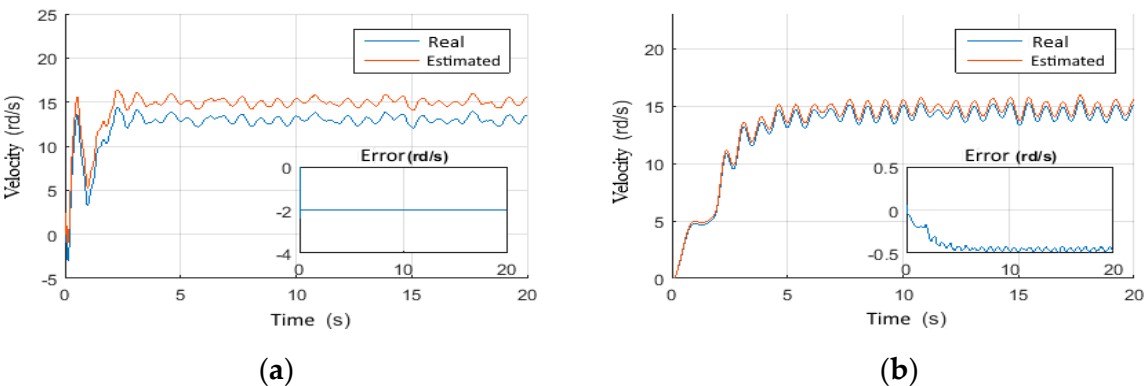

**Figure 42.** Real and estimated Top drive velocity under: (**a**) LQG controller, (**b**) H∞ controller.

However, it should be taken into consideration that since the H∞ observer-based controller proposed in this study was based on solving two cascaded Riccati equations, the choice of the covariance matrices $B_1$, $B_2$, and $D_{12}$ is very crucial for the approach convergence. Hence, it is highly recommendable to run many tests and scenarios with different covariance matrices values in the drilling field in order to guarantee that no matter what the top drive input signals are, the control system will converge to its required dynamic in the most robust way. Based on the obtained results in this research work, and in order to preserve the obtained robustness of the H∞ observer-based controller, we recommend the implementation of signal filters that can normalize and adapt the top drive input signals, namely the torque and the angular velocity, in an appropriate way similar to the simulation scenarios provided in this research work.

## 5. Conclusions

This work was devoted to the design of an observer-based controller in rotary drilling systems. The motivation behind this choice of control strategy includes the technical and practical difficulties that arise when measuring the down hole drilling torque and speed. For this reason, an observer with unknown input was inserted into the control loop. This observer with unknown input was capable of producing estimations of the real states of the drilling system, in addition to an estimate of a primordial quantity of the drilling process, which is the torque on bit. The approach adopted in this work to mitigate the high frequency stick-slip vibrations in a robust and efficient way in the drill-string of rotary drilling systems was the H∞ observer-based controller. A model of ten degrees of freedom of the drill string was developed based on the Simscape environment of Simulink/Matlab. Moreover, a model of the rock–bit interaction was established under the same environment along with the top drive model. To estimate the downhole states, especially the bit speed and the torque on bit, an observer was first designed and tested under different scenarios by comparing the measured and estimated states to ensure the reliability of the observer before designing and discussing the controller robustness and performance. The obtained results of the observer were very promising. Thus, the controller was implemented in cascade to ensure the tracking task. This controller's purpose was to drive the drill bits peed, relying upon the unknown in put observer outcomes in such a way to keep the bit speed steady even under extreme conditions of the weight on bit, model uncertainties, and measurement noises. That is why the obtained observer-based controller has been tested under different possible and probable scenarios from the drilling field. The obtained results have demonstrated that the controller robustness is very high, for which the high frequency stick-slip vibrations were suppressed in less than 5 s. This is a very short time in comparison to the manual suppression adopted nowadays in the drilling fields. Moreover, the severity of stick-slip vibrations has been quantitatively evaluated through the proposed empirical equation. Furthermore, the H∞ observer-based controller performance has been qualitatively compared to the LQG observer-based controller. The proposed approach demonstrated good performance even for an extreme case, in which it provided a rejection to unstructured perturbations unlike the LQG approach. Therefore, based on the obtained results, it is highly recommendable to implement the designed approach on an operating rotary drilling system towards the development of the so-called smart rotary drilling systems.

**Author Contributions:** Conceptualization, M.Z.D. and M.K.; methodology, M.Z.D. and S.D.; software, R.R.; validation, M.K., S.D. and M.Z.D.; formal analysis, R.R.; investigation, M.Z.D.; resources, M.K.; data curation, R.R. and S.D.; writing—original draft preparation, R.R. and M.Z.D.; writing—review and editing, S.D. and M.Z.D.; visualization, R.R.; supervision, M.Z.D. and S.D.; project administration, M.K. All authors have read and agreed to the published version of the manuscript.

**Funding:** This research received no external funding.

**Institutional Review Board Statement:** Not applicable.

**Informed Consent Statement:** Not applicable.

**Data Availability Statement:** There are no acknowledgments for this paper.

**Conflicts of Interest:** The authors declare no conflict of interest.

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
