# Peer review of "Observer-Based H∞ Controller Design for High Frequency Stick-Slip Vibrations Mitigation in Drill-String of Rotary Drilling Systems"

_vibration, doi:10.3390/vibration5020016_

Round 1

Reviewer 1 Report

I have attached file to this email.

Author Response

Revised paper: Manuscript ID: vibration-1646797

"Observer-Based H∞ Controller Design for High-Frequency Stick-Slip Vibrations Mitigation in Drill-String of Rotary Drilling Systems"

By Rami Riane, Mohamed Zinelabidine Doghmane, Madjid Kidouche, and Sofiane Djezzar *

We are pleased to submit herewith a revision of our paper based on the valuable comments of the reviewers. We would like to thank the reviewers for their constructive comments. The main changes are highlighted in the revised version in different colors where, comments for reviewer “1” are highlighted in yellow; those for reviewer “2” are highlighted in light green, while common comments are highlighted in Blue.

We have incorporated, in the presented revised version of this paper, all modifications required by taking into account each feedback/comment provided by all reviewers, as depicted in the rest of this letter. By the way, we have proceeded to a general check for English in order to improve the readability of our manuscript.

Finally, we would like to thank the reviewers for their constructive and important comments that have led to improving significantly our paper. We hope that the revised version will meet their expectations, and we remain available to clarify our answers and to treat any new comments or feedback from your side.

Yours sincerely,

  1. Riane, M. Z. Doghmane, M. Kidouche, and S. Djezzar (corresponding author).

Reviewer 2 Report

The authors present an interesting H-infinity based control design for mitigating stick-slip vibrations that occur in rotary drilling operations.  The results are promising and the findings could merit publication, if the following can be satisfactorily addressed:

i) The authors appear to be quite unaware of a significant body work related to the nonlinear dynamics and control of drill string oscillations with consideration of delay effects, loss of contact dynamics, and other aspects.  See the following for a partial list of recent studies: a) Nonlinear Oscillations of a Flexible Rotor with a Drill Bit: Stick-Slip and Delay Effects, Nonlinear
Dynamics, Vol. 72, pp. 61-77, 2013. b) Bifurcations in the axial torsional state-dependent model of rotary drilling, International Journal of Non-Linear Mechanics, Vol. 99, pp. 13-30, 2018. c) Effects of High Frequency Drive Speed Modulation on Rotor with Continuous Stator Contact, International
Journal of Mechanical Sciences, Vol. 131-132, pp. 559-571, 2017.  d) Advantages of an LQR controller for stick-slip and bit-bounce mitigation in an oilwell drill string, 2012 ASME IMECE, pp.1305-1313.  e) Spatio-Temporal Dynamics of a Drill String with Complex Time-Delay Effects: Bit Bounce and Stick-Slip Oscillations, International Journal of Mechanical Sciences, Vol. 170, 2020.  f) Nonlinear Instabilities and Control of Drill-String Stick-Slip Vibrations with Consideration of State-Dependent Delay, Journal of Sound and Vibration, Vol. 473, 2020. g) Analysis and control of stick-slip oscillations in drilling systems, IEEE Trans. Control Systems Technology, Vol. 24, pp. 1582-1593, 2016. 

ii) The parameter ranges in which the controller is shown to be effective does not appear to cover the region in which nonlinear instabilities can occur as discussed in the studies of part a).  Can the authors comments on the region of applicability of the current controller design?

iii) Given the distributed nature of the drill string, the reduced order model used in this work may not be appropriate for a real drill-string system.  Comments are in order.  Also, why is back emf not modeled in the motor model.

iv)  Can the authors comment on the choice of the states to be measured?  They clearly don't have full state feedback.  It would be useful to discuss observability and controllability to better understand the effectiveness of the controller.

Author Response

(The authors gave the same response as above.)

Round 2

Reviewer 1 Report

Corrections were well done

Author Response

Djezzar Sofiane (corresponding author)

University of North Dakota (UND),

Grand Forks, North Dakota, United States

E-mail: sdjezzar@undeerc.org

March 5th, 2022

Dear Ms. Dale Du,

Second Round of Revised paper: Manuscript ID: vibration-1646797

"Observer-Based H∞ Controller Design for High-Frequency Stick-Slip Vibrations Mitigation in Drill-String of Rotary Drilling Systems"

By Rami Riane, Mohamed Zinelabidine Doghmane, Madjid Kidouche, and Sofiane Djezzar *

We are pleased to submit herewith the second revision of our paper based on the valuable comments of the reviewers. We would like to thank the reviewers for their constructive comments. The main changes are highlighted in yellow for reviewer 2 comment.

We have incorporated, in the second revised version of this paper, all modifications required by taking into account each feedback/comment provided by all reviewers, as depicted in the rest of this letter. By the way, we have proceeded to a general check for English in order to improve the readability of our manuscript.

Finally, we would like to thank the reviewers for their constructive and important comments that have led to improving significantly our paper. We hope that the revised version will meet their expectations, and we remain available to clarify more our answers and to treat any new comments or feedback from your side.

Yours sincerely,

  1. Riane, M. Z. Doghmane, M. Kidouche, and S. Djezzar (corresponding author).

Reviewer 2 Report

The authors' revisions are appreciated.  However, it would be important to acknowledge the limitations of the work and discuss how they may be addressed in the closing section.

Author Response

(The authors gave the same response as above.)
